# CIRCLE-ROPE: CONE-LIKE DECOUPLED ROTARY POSITIONAL EMBEDDING FOR VISION-LANGUAGE MODELS

## ABSTRACT

Rotary Position Embedding (RoPE) is a widely adopted technique for encoding relative positional information in large language models (LLMs). However, when extended to vision-language models (VLMs), RoPE and its variants enforce relative positional dependencies separately within text and image tokens, introducing unintended cross-modal positional biases. For example, image tokens depicting semantically consistent content are assigned distinct positional encodings solely due to spatial location variations. As a result, such tokens exhibit entirely different relative positional relationships with their corresponding text tokens, ultimately leading to misaligned cross-modal representations. To address this, we propose Per-Token Distance, a simple yet effective metric for quantifying the independence of positional encodings across modalities. Informed by this analysis, we introduce Circle-RoPE, a novel encoding scheme designed to eliminate spurious cross-modal biases. Our key idea is to project image token indices onto a *ring* that is orthogonal to the linear axis of text token indices, thereby forming a cone-like structure in the positional encoding space. In this configuration, each text token (point on the linear text axis) becomes the apex of a cone and maintains an equal distance to all image tokens (points on the circular image *ring*), reducing artificial cross-modal biases while preserving intra-image spatial information. To further enhance performance, we propose a staggered strategy that applies different RoPE variants across layers. Extensive experiments demonstrate that our method effectively preserves spatial information from images while reducing relative positional bias, offering a more robust and flexible positional encoding framework for VLMs.

## 1 INTRODUCTION

In the rapidly evolving transformer landscape, Rotary Position Embedding (RoPE) [19] has emerged as the de facto standard for encoding relative positional information in large language models (LLMs). When extending models to handle both textual and visual inputs, as in Vision-Language Models (VLMs), a challenge emerges: how to effectively encode positional information across disparate modalities. Text is inherently sequential, while visual data is spatially structured, characterized by attributes such as location, orientation, viewpoint, and scale—properties that are fundamentally different and largely uncorrelated with textual order.

Different approaches have been explored to tackle this issue. For instance, Figure 1(a) illustrates models like LLaVA [13], Emu3 [21], InternLM-VL [4], and DeepSeek-VL2 [23], which flatten image tokens into a 1D sequence and concatenated them with text tokens, directly applying the standard 1D RoPE from LLMs to multimodal encoding. Figure 1(b) depicts the strategy used in mPLUG-Owl3 [27], where all image patches are simply assigned with the same positional index. Figure 1(c) depicts the positional encoding in M-RoPE [20] (Qwen2-VL), which preserves the spatial layout of images while modeling textual sequentiality, though it still concatenates image and text tokens in the same sequence as in Figure 1(a).

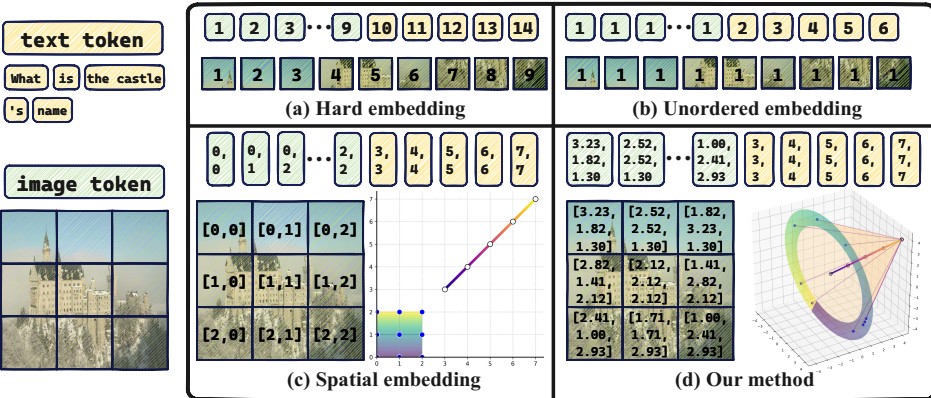

Figure 1: Text (yellow) and image (green) tokens are labeled with their position indices under different RoPE-based encoding schemes. (a) hard embedding method, which encodes image tokens by their flattened sequence; (b) unordered embedding method, assigning the same index to all image tokens within an image; (c) spatial embedding method, where image tokens are indexed according to their 2D positions in the original image; (d) our method, which remaps image token index onto a circle orthogonal to the text index direction, achieving a decoupled encoding.

All existing RoPE variants either flatten visual tokens into a 1D sequence or arrange them on a 2D grid before concatenating with text tokens. Both approaches, however, **introduce spurious cross-modal positional biases**—not from actual data relationships but from the hard-coded design of positional embeddings—which can undermine multimodal understanding. Figure 2 illustrates this issue with a visual question answering (VQA) example: "What type of religion is displayed high on the clock tower?" The phrase *high on* requires spatial reasoning, and *clock tower* requires object recognition. Yet their relationships to the correct image regions are distorted by index-based encoding. Two common biases emerge: (i) **semantic misalignment**—*high on* should align with the top of the tower (index 1) but is instead placed near index 8; and (ii) **inconsistent multi-token distances**—*clock tower* corresponds to multiple image tokens of the tower, but their relative distances to the text vary, leading to inconsistency.

In this work, we directly address the problem of positional bias by proposing **Circle Rotary Position Embedding (Circle-RoPE)**, a flexible positional encoding scheme that preserves intrinsic spatial relationships while maintaining consistent cross-modal alignment. At its core, our approach applies geometric transformations to the original coordinate indices of visual tokens before computing RoPE rotation factors. This ensures a fully decoupled encoding of text and image tokens, effectively mitigating cross-modal positional biases.

Specifically, we extend the M-RoPE mechanism, which represents image token indices by height–width coordinates, with two key innovations. First, we propose *Circular Image Token Index Projection* (**CIP**, Sec. 3.1) which projects 2D grid coordinates onto a circle in 3D space whose normal vector is aligned with the text vector. This transformation ensures orthogonal separation: each text token index lies along the normal vector and maintains equal Euclidean distance and consistent RoPE distances to all points on the circle, forming a cone-like structure. Meanwhile, relative spatial relationships among image tokens are preserved, as shown in Figure 1(d). This design effectively disentangles positional dependencies across modalities.

Second, we propose an *Alternating Geometry Encoding* (**AGE**, Sec. 3.2) strategy which cyclically switches between M-RoPE and our proposed Circle-RoPE across layers, leveraging their complementary strengths for more robust multimodal representations.

In summary, our contributions are twofold: (i) we identify and address cross-modal relative positional biases in existing RoPE variants through the design of Circle-RoPE; (ii) we validate its effectiveness across multiple LVMs and diverse multimodal tasks, achieving improved spatial consistency and visual reasoning.

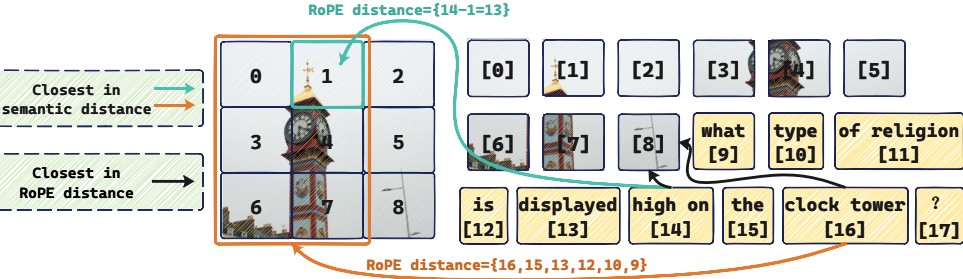

Figure 2: A VQA Example where image and text tokens are sequentially concatenated. The image token at index 8 exhibits the smallest RoPE distance to all text tokens, despite semantically closer image tokens being located elsewhere. The text token at index 16 exhibits varying distances to the six image patches that correspond to the same semantic content. These misalignments highlights how conventional RoPE methods introduce unintended relative positional biases.

## 2 PRELIMINARIES AND PROBLEM ANALYSIS

Recent work has extended RoPE from LLMs to multimodal settings, yet often overlooks a fundamental issue: the inherent misalignment between text token indices and image token positions. For example, while Qwen-VL's M-RoPE [20, 2] introduces 3D encoding for video (width, height, time) and improves performance; however, like other methods, it fails to decouple positional mappings across modalities. This failure forces unnatural relative position relationships between semantically related text and image tokens during RoPE encoding (as illustrated in Figure 2), introducing cross-modal bias in training and inference. Figure 1 shows some common approaches for implementing multimodal position embeddings:

- Hard embedding (Figure 1(a)): The image tokens are flattened into a 1D sequence and concatenated directly with the text tokens. While straightforward and intuitive, this method does not guarantee that each text token interacts independently with image tokens, often introducing unwanted positional biases instead of relying solely on high-level semantic understanding.

- Unordered embedding (Figure 1(b)): All image tokens are assigned the same index, thus the distance between any text token and all image tokens in the same image is identical. However, this approach ignores the relative positions containing spatial information among the image tokens themselves, leading to a loss of fine-grained visual structure.

- Spatial embedding (Figure 1(c)): Tokens are assigned 2D indices based on their positions in the image, providing more accurate spatial information among image tokens but still failing to guarantee independence between text and image token positions.

Existing approaches predominantly focus on encoding spatial information for images and sequential information for text independently, overlooking the potential interference caused by intertwined positional embeddings. This oversight can introduce unintended biases, distorting cross-modal alignment. Ideally, to eliminate such biases, the "distance" in RoPE index between each text token and all image tokens should remain consistent, ensuring positional independence across modalities.

Table 1: PTD values of different RoPE methods.

| Embedding method | Hard | Unordered | Spatial | Ours |
|---|---|---|---|---|
| Relative position information | ✔ | ✗ | ✔ | ✔ |
| PTD | 2.22 | 0 | 0.64 | **0** |

**Per-Token Distance (PTD) Metric.** To quantify and compare how different RoPE-based methods affect the relative position relationship between text and image tokens, we design a metric called *Per-Token Distance* (PTD). PTD evaluates the independence between the text token index and the image token index after the

application of positional encoding. Its formal definition is as follows: suppose the index list of image tokens is $I = \{i_1, i_2, ..., i_{N_{\text{image}}}\}$ with size $N_{\text{image}}$, and the index list of text tokens is $T = \{t_1, t_2, ..., t_{N_{\text{text}}}\}$ with size $N_{\text{text}}$. The PTD is calculated as:

$$\text{PTD} = \frac{1}{N_{\text{image}} N_{\text{text}}} \sum_{t \in T} \sum_{i \in I} \left| d(t,i) - \bar{D}_t \right|, \quad \bar{D}_t = \frac{1}{N_{\text{image}}} \sum_{i \in I} d(t,i) \tag{1}$$

where $d(x,y)$ denotes the Euclidean distance between $x$ and $y$. A smaller PTD value indicates a lower variance in the distances from each text token to the set of image tokens. This uniformity signals a higher degree of disentanglement between the text and image token indices. We compute PTD for three typical multimodal encoding methods, *i.e.*, hard embeeding (Figure 1(a)), unordered embedding (Figure 1(b)), and spatial embedding (Figure 1(c)). For convenience, we set $N_{\text{image}} = 9$ and $N_{\text{text}} = 5$. The PTD values are show in the Table 1. A non-zero PTD value after applying existing RoPE methods directly indicates the presence of cross-modal relative positional bias. This bias can hinder further performance improvements in VLMs.

Therefore, we propose to map all image token indices to positions equidistant from very text token index, aiming to minimize the PTD metric (ideally achieving 0) and mitigate cross-modal positional bias.

## 3 METHOD

We propose a novel positional encoding method for VLMs, **Circle Rotary Position Embedding (Circle-RoPE)**. Its core idea is to transform image token indices $(w, h)$ through a series of coordinate projections before applying the rotary matrix [20], thereby removing undesired cross-modal relative positional biases while preserving spatial relationships among iamge tokens. Circle-RoPE consists of two components: *Circular Image Token Index Projection* (**CIP**, Sec. 3.1) and *Alternating Geometry Encoding* (**AGE**, Sec. 3.2), with the details elaborated in the following sections.

### 3.1 CIRCULAR IMAGE TOKEN INDEX PROJECTION

We begin by designing Circular Image Token Index Projection (CIP) to fully decouple image token indices from text token indices, *i.e.*, achieve PTD = 0. The key idea of CIP is to project image token indices onto a structured geometric space, ensuring uniform RoPE distances to any text token and eliminating unintended positional biases. The CIP process consists of three key steps:

(i) *Coordinate Centralization*: Shift the geometric center of all image token indices to the origin, standardizing the coordinate reference.

(ii) *Mixed-Angle Circular Mapping*: Project the centralized image token indices onto a 2D circular trajectory. The angular position of each index is determined by a combination of its spatial-origin angle and its grid-index angle, with a defined radius for structural consistency.

(iii) *Target Plane Rotation*: Rotate the 2D circular structure from previous step onto a specific plane in 3D space. The orientation of this plane is determined by the text token indices, ensuring orthogonality between the image token index plane and the text token index direction.

In M-RoPE [20], image token indices are represented separately by width and height coordinates, text tokens use 1D positional index equivalent to standard RoPE. As show in Figure 3(a), given the original M-RoPE index, we obtain the image token index based on a regular grid, denoted as $C = \{(x_{ij}, y_{ij})\}_{i \in W, j \in H}$, where $W = \{0, 1, \ldots, w-1\}$ and $H = \{0, 1, \ldots, h-1\}$. Here, $w$ and $h$ correspond to the width and height of the image after tokenization. For clarity, we let $W$ correspond to the $x$-axis and $H$ to the $y$-axis. The goal of CIP is to transform the original image token index $C = \{(x_{ij}, y_{ij})\}$ into decoupled indices from the text tokens, resulting in $C_{\text{proj}} = \{(x_{ij}^{\text{proj}}, y_{ij}^{\text{proj}})\}$. These transformed indices are then directly used for RoPE computation.

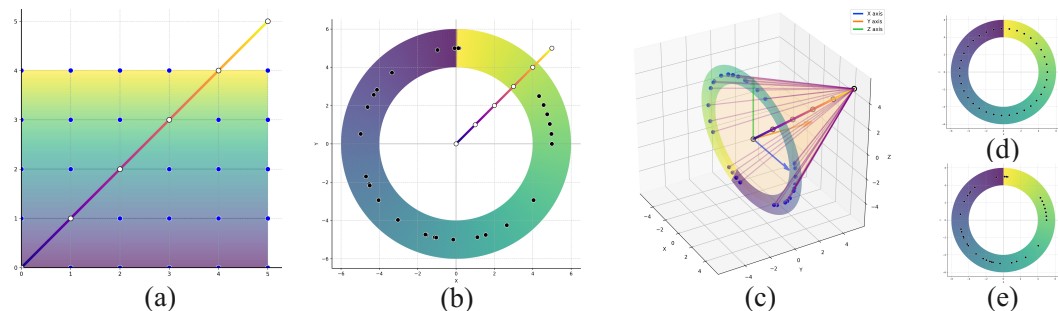

Figure 3: Transformation steps for *Circular Image Token Index Projection* (**CIP**): (i) coordinate centralization, (ii) mixed-angle circular mapping, and (iii) target plane rotation as described in Sec 3.1. For clarity, the starting points of text and image indices are aligned in above figure, preserving their relative positional distances without loss of generality. (a) Initial M-RoPE [20] index in step (i); (b) 2D circular structure after steps (i) and (ii); (c) 3D circular structure after step (iii); (d) Grid-index angle (GA) in step (ii); (e) Spatial-origin angle (SA) in step (ii).

### 3.1.1 COORDINATE CENTRALIZATION

To facilitate subsequent transformations, we first center the image token index coordinates. Specifically, the geometric center $P_{\text{center}} \in \mathbb{R}^2$ of the image token indices is calculated as follows:

$$P_{\text{center}} = \frac{1}{2}\left(\max_i(C_i) + \min_i(C_i)\right) \tag{2}$$

We then subtract this center point from all original coordinates to obtain the centered coordinates:

$$C' = C - P_{\text{center}} \tag{3}$$

This ensures that the geometric center of $C' = \{(x'_{ij}, y'_{ij})\}$ is located at the origin $(0,0)$, providing a natural reference frame for subsequent projection and rotation.

### 3.1.2 MIXED-ANGLE CIRCULAR MAPPING

To construct a cone-like structure that effectively decouples the text token indices from the image token indices, we first transform the centered image token coordinates $C'$ into polar coordinates and project them onto a 2D circle. During this transformation, the angular position of each point on the circle is determined by a combination of its spatial-origin angle (SA) and grid-index angle (GA), while the radius $R$ remains flexible. The resulting 2D circular structure is illustrated in Figure 3(b). We detail the calculation of these two angles and the radius in the following.

**Angle Calculation:** We combine two complementary angles to balance spatial structure with index information, determining the transformed angle for each image token index:

(1) *Spatial-Origin Angle $\theta_{ij}^{SA}$ (SA)*: we first compute the polar angle of each centered point $(x'_{ij}, y'_{ij})$:

$$\theta_{ij}^{\text{atan2}} = \text{atan2}(y'_{ij}, x'_{ij}) \tag{4}$$

where function $\text{atan2}(y, x)$ returns the angle between the point $(x, y)$ and the positive $x$-axis, in $(-\pi, \pi]$. Then, we normalize these angles to the range $[0, 2\pi)$:

$$\theta_{\min} = \min_{i,j}(\theta_{ij}^{\text{atan2}}), \quad \theta_{\max} = \max_{i,j}(\theta_{ij}^{\text{atan2}}), \quad \Delta\theta = \theta_{\max} - \theta_{\min} \tag{5}$$

thus, as illustrated in Figure 3(e), the SA is given by:

$$\theta_{ij}^{\text{SA}} = \begin{cases} \frac{\theta_{ij}^{\text{atan2}} - \theta_{\min}}{\Delta\theta} \times 2\pi & \text{if } \Delta\theta > 0 \\ 0 & \text{if } \Delta\theta \leq 0 \end{cases} \tag{6}$$

(2) *Grid-Index Angle $\theta_{ij}^{GA}$ (GA):* We flatten the $H \times W$ grid into a 1D sequence with $N = H \times W$ points, assigning each point a uniformly spaced angle based on its flattened index $k \in \{0, ..., N-1\}$:

$$\theta_k^{\text{GA}} = \frac{k}{N} \times 2\pi \tag{7}$$

mapping the index $k$ back to the grid position $(i, j)$ yields $\theta_{ij}^{\text{GA}}$, ensuring the angles are equally spaced around the circle, as shown in Figure 3(d).

(3) *Angle Mixing:* The final mixed angle $\theta_{ij}^{\text{mix}}$ is computed by a weighted average of the two strategies:

$$\theta_{ij}^{\text{mix}} = \alpha \cdot \theta_{ij}^{\text{SA}} + (1 - \alpha) \cdot \theta_{ij}^{\text{GA}} \tag{8}$$

the coefficient $\alpha \in [0, 1]$ controls the balance between preserving spatial information and enhancing the uniqueness of each position. While the SA retains more spatial structure, the GA leads to a clearer separation between positions, making it easier for the model to distinguish between them.

**Radius Calculation:** The choice of radius $R$ affects the scale of the transformed coordinates and influence the effective frequency range used by RoPE [19]. We provide two strategies here:

(1) Fixed: Use a predefined constant value $R_{\text{fix}}$.

(2) Automatic (auto-$k$): Scale $R$ based on a measure of the spread of the centered coordinates $C'$, such as the maximum $L_2$ norm:

$$R_{\text{auto}} = k \times \max_{i,j} \|(x'_{ij}, y'_{ij})\|_2 \tag{9}$$

where $k$ is a predefined scaling factor (*e.g.*, $k = 1$ or $k = 2$).

**Mapping to the Circle:** Based on the computed angle $\theta^{\text{mix}} ij$ and radius $R$, the new coordinates of each image token index on the $XY$-plane are given by $x_{ij}^{\text{circ}} = R\cos(\theta_{ij}^{\text{mix}})$ and $y_{ij}^{\text{circ}} = R\sin(\theta_{ij}^{\text{mix}})$, which collectively form a circle $C_{\text{circ}} = \{(x_{ij}^{\text{circ}}, y_{ij}^{\text{circ}})\}$, as illustrated in Figure 3(b).

### 3.1.3 TARGET PLANE ROTATION

After above transformation, visual token index points are mapped to $C_{\text{circ}}$ on the $XY$-plane. To decouple them from the text token index (*i.e.*, achieve PTD=0), we further rotate the circle in 3D space so that its plane is perpendicular to vector $V_{\text{text}}$ defined by the text token index, *i.e.*, $V_{\text{text}}$ serves as the normal vector of the circle. For computational convenience, we extend $C_{\text{circ}}$ to 3D space by initializing the third (z) coordinate to zero for all points. The specific conversion process is as follows:

(1) Define the target plane normal: normalize the $V_{\text{text}}$ to obtain a unit normal vector $\mathbf{n}$:

$$\mathbf{n} = \frac{V_{\text{text}}}{\|V_{\text{text}}\|_2} = (n_x, n_y, n_z) \tag{10}$$

(2) Construct an orthonormal basis for the target plane: then define two orthonormal vectors $\{\mathbf{u}, \mathbf{v}\}$ lying in the target plane and orthogonal to $\mathbf{n}$:

$$\mathbf{u}' = (-n_y, n_x, 0) \quad \mathbf{u} = \frac{\mathbf{u}'}{\|\mathbf{u}'\|_2}, \quad \mathbf{v} = \mathbf{n} \times \mathbf{u} \tag{11}$$

where $\mathbf{u}$ is a unit vector lying in the target plane and orthogonal to $\mathbf{n}$, while $\mathbf{v}$ is also orthogonal to both $\mathbf{n}$ and $\mathbf{u}$, ensuring that $\mathbf{u}, \mathbf{v}, \mathbf{n}$ forms a right-handed orthonormal basis.

(3) Coordinate transformation: for each point $P_{ij}^{\text{circ}} = (x_{ij}^{\text{circ}}, y_{ij}^{\text{circ}}, 0)$ on $C_{\text{circ}}$, compute its new coordinate on the target plane as a linear combination:

$$P_{ij}^{\text{proj}} = x_{ij}^{\text{circ}}\mathbf{u} + y_{ij}^{\text{circ}}\mathbf{v} \tag{12}$$

Applying this transformation to all points yields the final projected set $C_{\text{proj}} = \{(x_{ij}^{\text{proj}}, y_{ij}^{\text{proj}}, z_{ij}^{\text{proj}})\}$, as illustrated in in Figure 3(c). These points lie on a circle in 3D space, with its normal vector aligned to $V_{\text{text}}$, ensuring PTD=0 and preserving spatial information relative to the image.

### 3.2 ALTERNATING GEOMETRY ENCODING (AGE)

In Transformer-based LVLMs, different layers tend to capture distinct geometric patterns, where lower layers focus on local details and higher layers emphasize global structure. Therefore, we propose **Alternating Geometry Encoding (AGE)**, which cyclically switches between the M-RoPE [20] index and the Circle-RoPE index across different Transformer layers, allowing the model to capitalize on the complementary strengths of multiple geometric representations.

## 4 EXPERIMENT

In this section, we first introduce our model configuration and parameter details. We then compare our proposed method with mainstream models. Finally, we conduct ablation studies to validate the effectiveness of our approach and analyze the contributions of different components.

### 4.1 TRAINING SETTING

To evaluate the effectiveness of our method, we employ Qwen2.5-VL [2] and LLaVA [13] as baseline models for our experiments. The only modification introduced is in the implementation of the positional encoding method; all other configurations are retained from the baseline model. During training, we exclusively update the parameters of the LLM component while keeping the parameters of the Vision-Language projection layers and the Vision Encoder frozen. All experiments are conducted under a unified training setup. The complete set of hyperparameter configurations are provided in appendix's Table 7. For training, we randomly sample one-tenth of the MAmmoTH-VL Instruct dataset (**12M**) [8] and exclude all video data, resulting in a subset named MAmmoTH-VL-Sub (**1M**). Our experiments demonstrate that even with this reduced data size, our method achieves significant performance improvements compared to the baseline.

### 4.2 COMPARISON WITH OTHER MODELS

This section evaluates the performance of Circle-RoPE on a diverse range of datasets, benchmarking it against state-of-the-art models such as SAIL-VL [5], InternVL2.5 [4], Ovis2 [16], Phi-3.5-vision [1], and various scales of MiniCPM-V-2 [26] and Qwen2.5-VL [2].

To ensure a comprehensive and fair comparison of the open-source models listed in Table 2, we employed VLMEvalKit [6] to evaluate all models under a unified protocol. As we utilized a third-party open-source toolkit, and the version of GPT used for evaluation differs from those reported in the original papers of some models, the results presented in the table may not be entirely consistent with the official results.

### 4.3 EXPERIMENT ON CIRCULAR MAPPING

We conducted ablation studies on the parameters used in Circular Image Token Index Projection (CIP). To validate the effectiveness of angle mixing and to select the optimal radius, we designed a series of ablation

Table 2: Performance of VLM Instruct models and our method (improvement over Qwen2.5-VL shown in parentheses).

| Dataset | SAIL-VL [5] | InternVL2.5 [4] | Ovis2 [16] | MiniCPM [26] V-2 | MiniCPM [26] V-2.6 | Phi-3.5 [1] vision | Qwen2.5-VL [2] | Ours |
|---|---|---|---|---|---|---|---|---|
| | 2B | 4B | 2B | 2.8B | 8B | 4.2B | 3B | 3B |
| MMMU$_{val}$ [29] | 41.44 | 51.56 | 43.78 | 37.00 | 43.44 | 44.44 | 50.22 | **52.11** (+1.89) |
| MMMU-Pro$_{overall}$ [30] | 14.51 | 26.65 | 21.21 | 14.77 | 20.26 | 16.42 | 27.92 | **28.44** (+0.52) |
| MathVista$_{mini}$ [15] | 60.70 | 60.60 | **64.50** | 40.80 | 60.20 | 43.70 | 62.40 | 63.40 (+1.00) |
| MMStar [3] | 56.47 | 58.53 | **58.67** | 41.00 | 57.53 | 47.40 | 54.13 | 58.20 (+4.07) |
| AI2D [9] | 77.72 | 81.38 | **82.77** | 64.77 | 81.28 | 77.59 | 78.14 | 81.80 (+3.66) |
| RealWorldQA [25] | 63.01 | 64.97 | **67.06** | 55.03 | 65.62 | 53.99 | 65.75 | 66.54 (+0.79) |
| InfoVQA [17] | 62.86 | 72.27 | 71.65 | 40.20 | 64.86 | 35.18 | 77.25 | **77.42** (+0.17) |
| **Avg Score** | 53.82 | 59.42 | 58.52 | 41.94 | 56.17 | 45.53 | 59.40 | **61.13** (+1.73) |

experiments. Specifically, we varied the angle mixing parameter $\alpha$ and explored different strategies for calculating the radius. As shown in Table 3, the model achieves the most balanced performance when $\alpha = 0.5$ and the radius is set to 10.

Additionally, we provide results for the baseline model after supervised fine-tuning (SFT) on the MAmmoTH-VL-Sub (**1M**) dataset. This allows for a direct comparison of how different parameter configurations affect model performance under the same conditions.

## 4.4 EXPERIMENT ON ALTERNATING GEOMETRY ENCODING

To thoroughly assess the impact of utilizing different geometry encoding strategies across various model layers, we systematically designed and evaluated four distinct encoding configurations. Specifically, the strategies we explored include: (1) applying Circle-RoPE consistently in all layers, thereby maintaining a uniform encoding approach

Table 3: Performance comparison across different CIP configurations.

| $\alpha$ | Radius | MMMU$_{val}$ [29] | MMMU-Pro$_{overall}$ [30] | MMStar [3] | MathVista$_{mini}$ [15] | Avg Score |
|---|---|---|---|---|---|---|
| baseline | | 50.22 | 27.92 | 54.13 | 62.40 | 48.67 |
| $\alpha = 0$ | auto | **52.38** | 28.12 | 57.50 | 61.70 | 49.93 |
| $\alpha = 0$ | 5 | 51.32 | 29.01 | 58.32 | 62.40 | 50.26 |
| $\alpha = 0$ | 10 | 51.49 | **29.13** | **58.57** | 62.70 | 50.47 |
| $\alpha = 0.3$ | 10 | 52.05 | 28.50 | 58.22 | 63.30 | 50.52 |
| $\alpha = 0.5$ | 10 | 52.11 | 28.44 | 58.20 | **63.40** | **50.54** |
| $\alpha = 0.7$ | 10 | 52.03 | 28.39 | 58.13 | 62.90 | 50.36 |
| $\alpha = 1$ | 10 | 52.16 | 28.35 | 57.70 | **63.40** | 50.40 |
| $\alpha = 0.5$ | auto | 50.04 | 26.64 | 57.30 | 62.20 | 49.05 |

throughout the network; (2) adopting Circle-RoPE only in the upper layers, from layer 19 to 36; and (3) employing Circle-RoPE exclusively in the lower layers, specifically layers 1 through 18, to evaluate the impact of introducing relative position bias at different depths of the model.

We also include (4) implementing an Alternating Geometry Encoding strategy, in which Circle-RoPE and M-RoPE are alternated at every successive layer to maximize the complementary strengths of both encoding methods. As illustrated in Table 4, the experimental results clearly demonstrate that the alternating strategy achieves the most robust performance among all tested configurations. This finding confirms that alternating between the two encoding methods enables the model to leverage the strengths of both approaches simultaneously. This finding suggests that leveraging the unique advantages of both encoding methods at different stages of the model can lead to enhanced overall effectiveness and more expressive geometric representations.

Table 4: Performance comparison across different AGE configurations.

| Strategy | MMMU (val) | MMMU_Pro | MMStar | MathVista_MINI | AI2D_TEST | ChartQA_TEST | InfoVQA | Avg score |
|---|---|---|---|---|---|---|---|---|
| strategy 1 | 51.32 | 28.41 | 55.93 | **65.20** | 80.39 | **84.15** | 76.92 | 63.19 |
| strategy 2 | 52.66 | 28.51 | **59.87** | **65.20** | 79.81 | 81.96 | 76.87 | 63.55 |
| strategy 3 | **53.48** | **28.62** | 59.30 | 64.50 | 79.30 | 82.61 | 77.35 | 63.59 |
| strategy 4 | 52.11 | 28.44 | 58.20 | 63.40 | **81.80** | 84.12 | **77.42** | **63.64** |

## 4.5 GENERALIZABILITY VERIFICATION ON DIFFERENT ARCHITECTURES

To validate the generalizability of our proposed Circle-RoPE, we conducted a rigorous ablation study on LLaVA [13] with a distinct architecture from the one primarily used in our work. We selected Llava-onevision-qwen2-0.5b as the base model, and performed experiments on the MAmmoTH-VL-Sub dataset. This setup provides a robust testbed for evaluating the adaptability and effectiveness of our method.

We compared four variants of the model to isolate the impact of our contributions: **Llava [1D-RoPE] (base)**: The original Llava-onevision-qwen2-0.5b model, serving as a foundational reference. **Llava [M-RoPE]**: we replaced Llava's 1D-RoPE with M-RoPE from Qwen2.5-VL. **Llava [Circle-RoPE]**: Our proposed Circle-RoPE was integrated into the Llava architecture, replacing its original 1D-RoPE.

The experimental results are summarized in Table 5. Our proposed Circle_RoPE consistently outperforms all other variants across every metric.

Table 5: Ablation study on the Llava-0.5B model to verify the generalizability of Circle-RoPE. Our method achieves the best performance across all benchmarks, demonstrating its effectiveness on a different model architecture.

| Model | MMMU-val | MMMU_Pro-avg | MMStar | MathVistamini | Avg Score |
|---|---|---|---|---|---|
| Llava [1D-RoPE] | 32.22 | 12.92 | 37.07 | 35.70 | 29.48 |
| Llava [M-RoPE] | 32.59 | 12.81 | 37.18 | 35.40 | 29.50 |
| **Llava [Circle-RoPE]** | **32.77** | **13.21** | **37.22** | **36.10** | **29.83** |

As shown in Table 5, Circle-RoPR demonstrates strong performance, surpassing both the baseline model LLaVA and the version adapted with M-RoPE. This demonstrates that the benefits of Circle-RoPE are not confined to the Qwen-VL architecture but are generalizable to other LVLMs. For the experiments on the Llava model, we directly applied the optimal hyperparameters ($\alpha$ and $R$) discovered on Qwen2.5-VL without any architecture-specific tuning. The consistent performance gains prove that Circle-RoPE is a versatile and stable module that can be readily integrated into different models.

## 5 CONCLUSION

In this paper, we address the challenges of directly applying RoPE to multimodal VLM settings. Existing methods primarily focus on extending RoPE to the vision modality while neglecting the critical interplay between the positional indices of vision and text tokens. To evaluate this overlooked aspect, we first introduce the per-token distance metric to quantify the misalignment. Building on these insights, we propose Circle-RoPE, a novel framework consisting of three transformation steps. Our key idea is to preserve the relative positional information within the vision modality while simultaneously mitigating erroneous relative position biases between text and image tokens. This decoupled positional encoding enhances cross-modal alignment, paving the way for more robust multimodal understanding.

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

APPENDIX

This appendix includes further analysis and discussion, related work, the hyperparameters adopted in our experiments, and pseudocode implementations.

## A   FURTHER ANALYSIS AND DISCUSSION

### A.1   THE ADAPTATION COST OF INTRODUCING CIRCLE-RoPE

We instantiate Circle-RoPE on the architecturally closest backbone, *Qwen2.5-VL*, and monitor step-wise training dynamics under SFT. We observed that even minor architectural modifications—such as altering the positional encoding—require substantial retraining with large-scale data for the model to adapt to the new positional distribution. We refer to this phenomenon as the adaptation cost.

Table 6: Step-wise training dynamics illustrating the *adaptation cost* when introducing Circle-RoPE on Qwen2.5-VL under SFT. At 3k steps, Circle-RoPE lags slightly behind; after ∼8.5k steps it surpasses the baseline on both benchmarks. Best per column is in **bold**.

| Model | Step | Loss ↓ | MMStar ↑ | MathVision ↑ |
|---|---|---|---|---|
| Qwen2.5-VL (SFT) | 3000 | 0.7997 | 57.94 | 20.16 |
| Circle-RoPE (SFT) | 3000 | 0.8077 | 57.53 | 20.13 |
| Qwen2.5-VL (SFT) | 8463 | **0.7666** | 58.07 | 20.56 |
| Circle-RoPE (SFT) | 8463 | 0.7725 | **58.20** | **20.95** |

Even on the most similar backbone, Circle-RoPE exhibits a measurable *adaptation cost*: at early training (3k steps) its performance is slightly below the SFT baseline (Table 6). With continued optimization (∼8.5k steps), the advantages emerge and eventually surpass the baseline on both MMStar ( +0.13) and MathVision ( +0.39). This indicates that even minor positional-encoding changes require non-trivial optimization to re-stabilize the representation geometry. Under limited compute and a relatively small SFT set, these gains are conservative rather than inflated. Choosing Qwen2.5-VL was thus the most pragmatic and reliable validation setting given our constraints; adopting a more dissimilar backbone would likely incur a larger adaptation cost that is computationally prohibitive. The fact that Circle-RoPE achieves improvements despite the initial dip and limited data provides evidence of robustness and headroom; we expect further gains with larger-scale pre-training or extended SFT schedules.

### A.2   EFFECTIVENESS OF ALTERNATING GEOMETRY ENCODING (AGE)

We introduce Alternating Geometry Encoding (AGE) into our method primarily for the following reasons:

**(1) Complementary strengths and preservation of spatial information.** While Circle-RoPE achieves image–text decoupling, it inevitably alters the strong grid-based spatial prior of image patches provided by the original RoPE. By alternating the two encoding methods, the model benefits from both: it reduces cross-modal positional bias (from Circle-RoPE) and fully utilizes the fine-grained internal spatial structure of the image (from RoPE), achieving a "1+1>2" effect.

**(2) Compatibility with pre-trained knowledge and smooth transition.** Our models are fine-tuned from Qwen2.5-VL, whose weights are deeply adapted to the original RoPE. Compared with applying a completely new encoding scheme to one contiguous part of the network, an alternating strategy minimizes the "shock" to the existing weight distribution. This enables smoother and more data-efficient convergence under limited SFT data, better integrating the pre-trained knowledge with the new capabilities introduced by Circle-RoPE.

In summary, AGE serves as an optional but effective mechanism that (i) fuses complementary geometric biases to preserve spatial reasoning while reducing cross-modal positional bias, and (ii) eases optimization by providing a gentler transition from RoPE-adapted weights to Circle-RoPE-enhanced representations. Empirically, our ablations reflect these stability and performance benefits.

### A.3 ENCODING TEMPORAL ORDER IN MULTI-IMAGE SEQUENCES

When the input contains multiple images, we explicitly encode their sequential order by translating each image's circular-encoding center along a fixed global axis. Concretely, let $c_i$ denote the center of the circular positional encoding for the $i$-th image in the sequence (indexed from $i{=}1$). We define a constant direction vector $g = [1, 1, 1]^\top$ and a stride $\Delta{=}1$ (default), and set

$$c_i^{\text{final}} \; = \; c_i \; + \; (i - 1)\, \Delta\, g.$$

This translation assigns each image a unique location in the 3D positional space while keeping the within-image geometric structure determined by Circle-RoPE intact.

For example, when we have a sequence with three images image1, image2, image3 whose original centers are at 0, the final centers become

$$c_1^{\text{final}} = 0 + [0, 0, 0], \quad c_2^{\text{final}} = 0 + [1, 1, 1], \quad c_3^{\text{final}} = 0 + [2, 2, 2].$$

## B RELATED WORK

VLMs unify visual and textual representations within a single transformer, yet effectively integrating modality-specific positional encodings remains a fundamental challenge. A common strategy is to apply RoPE [19] uniformly across the combined token sequence. However, its naive application to concatenated image and text tokens introduces cross-modal positional bias: the attention becomes sensitive to arbitrary positional offsets between the two modalities. This bias distorts cross-modal alignment, particularly since visual tokens often reside in distant segments of the sequence from relevant text, resulting in impaired information fusion.

Recent advances in multimodal LLMs and pixel-level understanding highlight the significance of unified architectures and position encoding strategies [7, 31, 10, 28, 11, 22]. Many VLMs, *e.g.*, Emu3 [21], InternLM-VL [4], Baichuan-Omni [12], Eve [18], DeepSeek-VL2 [24], and LLaVA series [13, 14], adopt a simple strategy of flattening all tokens in a 1D sequence and using shared position encoding such as RoPE for both text and image tokens.

A distinct research direction assigns a shared positional index to all visual tokens. For example, mPLUG-Owl3 [27] assigns all patches of an image the same position index (via a placeholder token) when applying RoPE. This interleaved scheme preserves the image's insertion position in the text context and reduces index disparity among image patches, alleviating some bias due to modality mixing.

A third strategy is to introduce spatial positional embeddings tailored to the 2D structure of images. Qwen2-VL [20] exemplifies this by decomposing RoPE into separate dimensions (height, width, and temporal indices) for images, i.e., Multimodal RoPE (M-RoPE). This approach encodes image patches with 2D coordinates instead of large 1D indices, thereby better aligning visual tokens with textual positions.

Each method partially mitigates cross-modal positional issues, yet none completely eliminates bias: shared-index approaches discard intra-image spatial structure, while both flattened 1D sequences and spatial embeddings may retain subtle cross-modal misalignment.

## C  HYPERPARAMETERS

Table 7: Training Hyperparameter Configuration for our method.

| Hyperparameter | Value |
| --- | --- |
| Base Model | Qwen2.5-VL-3B |
| Image Resolution | 512×512 |
| Global Batch Size | 128 |
| Learning Rate | 1e-6 |
| Optimizer | AdamW |
| LR Schedule | Cosine Decay |
| Number of Epochs | 1 |
| Warmup Ratio | 0.1 |
| Max Sequence Length | 4096 |

## D  PSEUDOCODE IMPLEMENTATION OF CIRCLE-ROPE

```python
import torch

def circular_image_token_projection(C: torch.Tensor, alpha: float, R: float, V_text: torch.Tensor):
    """
    Circular Image Token Projection in PyTorch style.

    Args:
        C (torch.Tensor): Original image token grid coordinates (N, 2).
        alpha (float): Angle mixing weight.
        R (float): Circle radius.
        V_text (torch.Tensor): Text vector direction, shape (3,).

    Returns:
        torch.Tensor: Projected coordinates (N, 3).
    """

    # ========================================================
    # Step 1: Coordinate Centralization
    # ========================================================
    P_center = 0.5 * (C.max(dim=0).values + C.min(dim=0).values)   # (2,)
    C_prime = C - P_center                                         # (N, 2)

    # ========================================================
    # Step 2: Mixed-Angle Circular Mapping
    # ========================================================

    # 2a. Calculate Spatial-Origin Angle (SA)
    raw_angles = torch.atan2(C_prime[:, 1], C_prime[:, 0])         # (N,)
    min_angle = raw_angles.min()
    max_angle = raw_angles.max()
    delta_theta = max_angle - min_angle

    if delta_theta > 0:
```

```
34          theta_SA = (raw_angles - min_angle) / delta_theta * 2 * torch.pi
35      else:
36          theta_SA = torch.zeros_like(raw_angles)
37
38      # 2b. Calculate Grid-Index Angle (GA)
39      N = C.shape[0]
40      k = torch.arange(N, device=C.device)                        # (N,)
41      theta_GA = (k.float() / N) * 2 * torch.pi
42
43      # 2c. Mix Angles
44      theta_mix = alpha * theta_SA + (1 - alpha) * theta_GA
45
46      # 2d. Map to 2D circle and expand to 3D
47      x_circ = R * torch.cos(theta_mix)
48      y_circ = R * torch.sin(theta_mix)
49      C_circ = torch.stack([x_circ, y_circ, torch.zeros_like(x_circ)], dim=-1)  # (N, 3)
50
51      # ========================================================
52      # Step 3: Target Plane Rotation
53      # ========================================================
54
55      # 3a. Construct orthonormal basis from text vector
56      n = V_text / V_text.norm()                              # (3,)
57      u_prime = torch.tensor([-n[1], n[0], 0.0], device=C.device)
58      if u_prime.norm() < 1e-6:
59          u_prime = torch.tensor([1.0, 0.0, 0.0], device=C.device)
60      u = u_prime / u_prime.norm()
61      v = torch.cross(n, u)
62
63      # 3b. Project points from 2D circle to 3D target plane
64      #     This is a linear combination of basis vectors u and v.
65      C_proj = C_circ[:, 0].unsqueeze(-1) * u + C_circ[:, 1].unsqueeze(-1) * v  # (N, 3)
66
67      return C_proj
```

# E   VISUALIZATION OF ATTENTION MAP

To further evaluate the impact of our proposed method, we provide the visualization of attention distributions. The proposed methodology enables the visualization of cross-modal attention for Circle-RoPE and Qwen2.5-VL-3B-Instruct [2], with evaluations performed on the $\text{MMMU}_{\text{test}}$ benchmark [29]. Concretely, we first isolate and extract the attention matrix from the final decoder layer. The average attention from all text tokens to their corresponding image regions is then computed, projected back to the image domain, and reconstructed into a coarse-grained grid. This grid is subsequently transformed into a heatmap, followed by smoothing and enlargement through bilinear interpolation. Finally, a power-law contrast enhancement is applied to highlight salient points. The visualization results show that our method is able to concentrate more effectively on the regions relevant to the given question while exhibiting fewer attentional allocations to irrelevant areas.

**Question:**
Salvador Manufacturing builds and sells snowboards, skis and poles. The sales price and variable cost for each are shown: Their sales mix is reflected in the ratio 7:3:2. What is the overall unit contribution margin for Salvador with their current product mix?

A.$3,540
B.$1,190
C.$1,905
D.$1,635

Answer with the option's letter from the given choices directly.

**Answer: D**

| Product | Sellings price per unit | Variable cost per unit |
|---|---|---|
| Snowboards | $320.00 | $170.00 |
| Skis | $400.00 | $225.00 |
| Poles | $ 50.00 | $ 20.00 |

| Product | Sellings price per unit | Variable cost per unit |
|---|---|---|
| Snowboards | $320.00 | $170.00 |
| Skis | $400.00 | $225.00 |
| Poles | $ 50.00 | $ 20.00 |

Ours

| Product | Sellings price per unit | Variable cost per unit |
|---|---|---|
| Snowboards | $320.00 | $170.00 |
| Skis | $400.00 | $225.00 |
| Poles | $ 50.00 | $ 20.00 |

Qwen2.5-VL-3B-Instruct

**Question:**
Given that points A and B on the ground are 80m apart (as shown in the diagram below), the level instrument is set up at the midpoint of AB. The height difference $h_{AB}=+0.228m$. When the level instrument is moved 3m away from point A, the reading on the leveling staff at point A is a' = 1.695m, and the reading on the leveling staff at point B is b' = 1.446m. Find the value of i.

A.i"=55.2"
B.i"=56.0"
C.i"=56.1"
D.i"=56.2"

Answer with the option's letter from the given choices directly.

**Answer: D**

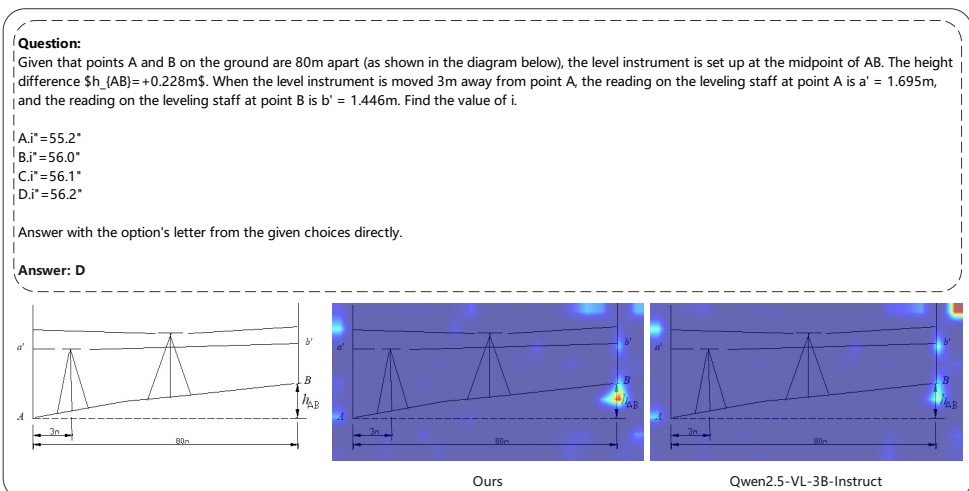

Ours                                    Qwen2.5-VL-3B-Instruct

