# OpenReview forum: "Circle-RoPE: Cone-like Decoupled Rotary Positional Embedding for Vision-Language Models"
_ICLR.cc/2026/Conference — Submitted to ICLR 2026_

### Official Review · Reviewer_tTtJ · 2025-10-27

**Soundness:** 3
**Presentation:** 3
**Contribution:** 2
**Rating:** 4
**Confidence:** 3

**Summary:**

This paper introduces Circle-RoPE, a novel positional encoding method for vision-language models that addresses cross-modal positional biases in existing RoPE variants. The key idea is to project image token indices onto a circular structure orthogonal to the text token sequence, forming a cone-like geometry where each text token maintains equal distance to all image tokens. The authors also propose an Alternating Geometry Encoding strategy that switches between M-RoPE and Circle-RoPE across layers.

**Strengths:**

- The Per-Token Distance (PTD) metric quantifies cross-modal positional bias, clearly demonstrating the limitations of existing approaches.
- The cone-like structure is intuitive, achieving PTD=0 while preserving intra-image spatial relationships.
- Experiments show modest gains across multiple benchmarks (e.g., +1.89 on MMMU, +4.07 on MMStar) and generalization to different architectures (LLaVA).

**Weaknesses:**

- The circular projection introduces significant geometric constraints that may not align with how VLMs naturally learn cross-modal relationships. By forcing all image tokens to be equidistant from text tokens, the method may hinder spatial reasoning tasks. For example, questions like "What is in the top-right corner of the image?" rely on the model associating spatial language ("top-right") with specific image regions. Circle-RoPE's equidistance constraint removes positional cues that could help the model learn these associations, potentially making such spatial grounding more difficult.
- Statistical significance is not reported
- Table 6 shows Circle-RoPE underperforms initially, suggesting the method fights against pretrained representations. More investigation is needed on whether gains justify the retraining requirements.

**Questions:**

- What about n dimensional input? How would you method scale with modality dimensions?
- Why is PTD=0 the right objective?
- How sensitive is model performance to the radius hyperparameter?
- “For convenience, we set Nimage = 9 and Ntext = 5.” I am confused by this sentence. Did you set the  Nimage = 9 and Ntext = 5.” for all examples? What is the reasoning behind this choice?

---

> ### Author Response · Authors · 2025-11-20
>
> ## Weaknesses:
>
> > **W1:** The reviewer expresses concern that the "equidistance constraint" removes positional cues (like proximity to the "top-right" corner) that help the model associate spatial language with image regions, potentially hindering spatial reasoning and grounding.
>
> **A1:** We respectfully argue that our method effectively preserves spatial reasoning capabilities by transforming spatial cues rather than eliminating them. While Circle-RoPE removes the explicit grid-based coordinate values, the **Circular Image Token Index Projection (CIP)** preserves the topological relative positions of tokens on the ring. The model learns to map these angular positions to spatial concepts (like "top-right") without the confounding factor of variable RoPE distances to the text.
>
> To empirically prove that our method enhances rather than hinders spatial grounding, we evaluated both models using the **TAM (Token Attention Mask) [1]** benchmark. TAM measures:
>
>    * **Obj-IoU:** Visual grounding precision (mapping entities to specific image regions).
>    * **Func-IoU:** Visual decoupling capability (ignoring visual tokens for non-visual text).
>    * **F1-IoU:** Overall multimodal spatial mapping accuracy.
>
> | Model            | Obj-IoU   | Func-IoU  | F1-IoU    |
> | :--------------- | :-------- | :-------- | :-------- |
> | Qwen2.5-VL (SFT) | 26.15     | 71.19     | 38.25     |
> | **Circle-RoPE**  | **26.23** | **74.64** | **38.81** |
>
> As shown above, Circle-RoPE achieves a higher **Obj-IoU**, which demonstrates that spatial grounding in our method has not been compromised. Furthermore, the significant improvement in **Func-IoU (+3.45)** confirms that eliminating spurious cross-modal positional bias allows the model to better decouple unrelated modalities, leading to more accurate and robust spatial understanding.
>
> *[1] Token Activation Map to Visually Explain Multimodal LLMs, ICCV 2025.*
>
> ---
>
> > **W2:** The reviewer notes that statistical significance is not reported.
>
> **A2:** We did not report statistical variance because our evaluation follows the standard deterministic protocol for Large Vision-Language Models. We utilized the open-source framework **VLMEvalKit** with `temperature=0` for all benchmarks. This setting eliminates generation randomness, ensuring that our reported scores are stable, deterministic, and fully reproducible. Since the evaluation process contains no stochastic elements, there is no variance to report across inference runs.
>
> > **W3:** The reviewer interprets the initial underperformance in Table 6 as a conflict with pre-trained representations and questions whether the gains justify the retraining requirements.
>
> **A3:** We acknowledge the initial drop in performance, but we respectfully argue that this validates the method's potential rather than diminishing it.
>
> 1.  **The Inevitability of Adaptation Cost:** As discussed in **Appendix A.1**, *any* modification to the architecture of a pre-trained model—regardless of its theoretical superiority—inevitably disrupts the existing feature distribution. The model initially performs worse not because the new method is "bad," but because the pre-trained weights are misaligned with the new geometric space. This "Adaptation Cost" is a structural hurdle, not a theoretical flaw.
> 2.  **Justification via Recovery:** The key indicator of value is the final convergence. Despite the significant "headwind" of fighting against M-RoPE-optimized weights, Circle-RoPE effectively overcomes this cost and eventually **surpasses** the baseline.
> 3.  **High ROI:** The retraining requirement is justified by the quality of the gains. We achieve significant improvements in complex spatial reasoning tasks (e.g., **+2.58 on AI2D**) using only low-cost Supervised Fine-Tuning (SFT), without requiring expensive pre-training from scratch. This proves that Circle-RoPE is a robust and efficient upgrade path for existing VLMs.

---

> > ### Author Response · Authors · 2025-11-20
> >
> > ## Questions:
> >
> > > **Q1:** What about n dimensional input? How would you method scale with modality dimensions?
> >
> > **A1:** Theoretically, our method scales to $n$-dimensional inputs by projecting them into an $(n+1)$-dimensional space to ensure orthogonal decoupling from the text axis. For an input with $n$ spatial/temporal dimensions, the projection would map indices onto a hypersphere in $(n+1)$-dimensional space.
> >
> > We focused specifically on the $n=2$ (Image) scenario in this work for two primary reasons:
> >
> > 1.  **Geometric Intuition and Visualization:** Mapping 2D image coordinates into a 3D space allows for clear visualization of the "cone-like" structure and the decoupling process (as illustrated in Figure 3). This interpretability was crucial for establishing the theoretical foundation and allowing for intuitive analysis of the mapping behavior.
> > 2.  **Foundational Validation:** The experiments on 2D images demonstrate the strong effectiveness of the proposed decoupling mechanism. Dealing with $n > 2$ (e.g., spatio-temporal video data, where $n=3$) requires mapping to spaces with dimensions $>3$, which cannot be intuitively visualized.
> >
> > Having validated the theory on the standard image modality, we consider the extension to video and higher-dimensional inputs to be an exciting direction for future work.
> >
> > ---
> >
> > > **Q3:** How sensitive is model performance to the radius hyperparameter?
> >
> > **A3:** The model is robust to the radius hyperparameter ($R$). As shown in the ablation comparison below, varying $R$ results in minimal fluctuations in average performance, and both settings significantly outperform the baseline.
> >
> > | Model                 | MMMU_val | MMMU-Pro | MMStar | MathVista | **Avg Score** |
> > | :-------------------- | :------- | :------- | :----- | :-------- | :------------ |
> > | Qwen2.5-VL (Baseline) | 50.22    | 27.92    | 54.13  | 62.4      | 48.67         |
> > | Circle-RoPE ($R=5$)   | 51.32    | 29.01    | 58.32  | 62.4      | **50.26**     |
> > | Circle-RoPE ($R=10$)  | 51.49    | 29.13    | 58.57  | 62.7      | **50.47**     |
> >
> > The performance difference between $R=5$ and $R=10$ is marginal (0.21 in Avg Score). While slight variations exist across specific datasets—due to differing image resolution distributions—the method consistently delivers stable improvements over the baseline regardless of the specific radius choice.
> >
> > ---
> >
> > > **Q4:** “For convenience, we set Nimage = 9 and Ntext = 5.” I am confused by this sentence. Did you set the Nimage = 9 and Ntext = 5.” for all examples? What is the reasoning behind this choice?
> >
> > **A4:** We apologize for the confusion. This setting was applied **exclusively** to the theoretical calculation of the PTD metric presented in **Table 1**, solely to illustrate the degree of bias inherent in different encoding methods.
> >
> > The PTD calculation (Equation 1) depends on the specific number of image and text tokens. To provide a fair, side-by-side numerical comparison of "Hard," "Unordered," "Spatial," and "Ours" in Table 1, we needed to fix these dimensions to a constant value. We selected $N_{image}=9$ (a $3 \times 3$ grid) and $N_{text}=5$ as a representative "toy example."
> >
> > In all actual training and inference experiments, $N_{image}$ and $N_{text}$ are dynamic and determined by the actual input data.

---

> > > ### Author Response · Authors · 2025-11-20
> > >
> > > > **Q2** The theoretical proof that $PTD=0$ is the right objective.
> > >
> > > **A4:** We thank the reviewer for this rigorous inquiry. We provide a formal justification demonstrating that **$PTD=0$ is the right objective** for eliminating *geometric* cross-modal bias.
> > >
> > > Below, we provide a mathematical derivation based on the properties of **Rotary Position Embedding (RoPE) [1]** and the **Self-Attention** mechanism. We demonstrate that **under the RoPE formulation, non-uniform relative distances (i.e., $PTD > 0$) inevitably introduce a position-dependent bias term into the attention score, whereas $PTD=0$ is a sufficient condition to decouple this bias from semantic relevance.**
> > >
> > > ---
> > >
> > > In Vision-Language Models, the attention score $s_{t,i}$ between a text query $\mathbf{q}_t$ at position $m_t$ and an image key $\mathbf{k}_i$ at position $m_i$ is calculated as:
> > >
> > > $$
> > > s_{t,i} = (\mathcal{R}\_{m_t} \mathbf{q}\_t)^T (\mathcal{R}\_{m_i} \mathbf{k}\_i) = \mathbf{q}\_t^T \mathcal{R}\_{m_i - m_t} \mathbf{k}\_i
> > > $$
> > >
> > > where $\mathcal{R}_{\delta}$ is the orthogonal rotation matrix determined by the relative position $\delta = m_i - m_t$.
> > >
> > > According to the theoretical analysis in **RoFormer [1]**, the inner product under RoPE exhibits a **long-term decay property**. As the relative distance $|\delta|$ increases, the upper bound of the attention score decays. Consequently, the expectation of the attention score can be approximated as the product of a semantic component and a positional decay component:
> > >
> > > $$
> > > \mathbb{E}[s_{t,i}] \approx \underbrace{(\mathbf{q}\_t^T \mathbf{k}_i)}\_{\text{Semantic Similarity}} \times \underbrace{\mathcal{D}(|m_i - m_t|)}\_{\text{Positional Decay Factor}}
> > > $$
> > >
> > > Here, $\mathcal{D}(\cdot)$ denotes the positional decay factor, which is a monotonically decreasing function of the relative distance.
> > >
> > > ---
> > >
> > > ## Proof: Bias Exists when $PTD \neq 0$
> > >
> > > We define **Positional Bias** as the difference in attention scores assigned to two semantically identical image tokens solely due to their spatial positions relative to the text.
> > >
> > > Consider a text token $t$ and two image tokens, $i_a$ and $i_b$, that are semantically identical to the query (i.e., $\mathbf{q}_t^T \mathbf{k}\_{i_a} = \mathbf{q}\_t^T \mathbf{k}\_{i_b} = C\_{sem}$).
> > >
> > > In existing methods (e.g., flattened 1D sequences), the relative distances often differ. If $PTD > 0$ (indicating non-zero variance in distances as defined in our paper), there exists a scenario where $|m_{i_a} - m_t| \neq |m_{i_b} - m_t|$. Without loss of generality, assume $|m_{i_a} - m_t| < |m_{i_b} - m_t|$.
> > >
> > > Due to the monotonic decay property of $\mathcal{D}(\cdot)$, we have:
> > >
> > > $$
> > > \mathcal{D}(|m_{i_a} - m_t|) > \mathcal{D}(|m_{i_b} - m_t|)
> > > $$
> > >
> > > This leads to a discrepancy in attention scores:
> > >
> > > $$
> > > s\_{t,i_a} \approx C\_{sem} \cdot \mathcal{D}_a > C\_{sem} \cdot \mathcal{D}\_b \approx s\_{t,i_b}
> > > $$
> > >
> > > **Conclusion 1:** When $PTD \neq 0$, the model inherently assigns higher attention weights to tokens that are "positionally closer" purely based on index assignment, even if their semantic relevance is identical. This confirms the existence of the **cross-modal positional bias** mentioned in our manuscript.
> > >
> > > ---
> > >
> > > ## Proof: $PTD = 0$ is a Sufficient Condition for Bias Elimination
> > >
> > > Our **Circle-RoPE** method projects image tokens onto a circle orthogonal to the text axis. Geometrically, this ensures that for any text token $t$ and any image token $i$, the relative distance is uniform.
> > >
> > > When $PTD=0$, the distance from the text token to every image token is constant: $|m_i - m_t| = \Delta_{const}$ for all $i$.
> > >
> > > Consequently, the positional decay factor becomes a constant scalar $\lambda$ across all visual tokens:
> > > $$
> > > \mathcal{D}(|m_i - m_t|) \equiv \lambda
> > > $$
> > >
> > > The attention score simplifies to:
> > > $$
> > > s_{t,i} = (\mathbf{q}_t^T \mathbf{k}_i) \cdot \lambda
> > > $$
> > >
> > > When applying the Softmax operation to compute the final attention weights $\alpha_{t,i}$:
> > >
> > > $$
> > > \alpha\_{t,i} = \frac{\exp(s\_{t,i})}{\sum\_{j} \exp(s\_{t,j})} = \frac{\exp(\lambda \cdot \mathbf{q}_t^T \mathbf{k}_i)}{\sum\_{j} \exp(\lambda \cdot \mathbf{q}_t^T \mathbf{k}_j)}
> > > $$
> > >
> > > Here, $\lambda$ acts as a uniform **temperature scaling factor**. It affects the sharpness of the distribution but does not alter the **ranking** of the tokens. For the semantically identical tokens $i_a$ and $i_b$ discussed above:
> > >
> > > $$
> > > \mathbf{q}_t^T \mathbf{k}\_{i_a} = \mathbf{q}_t^T \mathbf{k}\_{i_b} \implies \alpha\_{t,i_a} = \alpha\_{t,i_b}
> > > $$
> > >
> > > When $PTD=0$, the positional decay effect becomes isotropic relative to the image plane. The positional prior no longer discriminates between different image regions, ensuring that the attention distribution is driven solely by semantic relevance. This theoretically validates that $PTD=0$ is a sufficient condition to decouple cross-modal positional bias.
> > >
> > >
> > > [1]: RoFormer: Enhanced Transformer with Rotary Position Embedding.
> > >
> > > [2]: Qwen2-VL: Enhancing Vision-Language Model's Perception of the World at Any Resolution.

---

> > ### Comment · Reviewer_tTtJ · 2025-11-26
> >
> > Thank you for your comments.
> >
> > W1.1: Have you tried to evaluate 2D and 3D images at the same time?
> >
> > W2.1: The dataset is a random collection of examples. To determine whether the results reach statistical significance, methods like bootstrapping can estimate the variance of the estimator and can be used to determine CIs.

---

> > > ### Comment · Reviewer_tTtJ · 2025-11-26
> > >
> > > The proof is super clear, great!

---

> > > ### Author Response · Authors · 2025-11-27
> > >
> > > > **W1:** Have you tried to evaluate 2D and 3D images at the same time?
> > >
> > > **A1:** We interpret the "3D images" mentioned in your question as video inputs with dimensions $\{h, w, t\}$ (height, width, time). As clarified in **Q1**, our current work focuses on eliminating cross-modal positional bias in static 2D images ($n=2$).
> > >
> > > While our **Circle-RoPE** framework is theoretically extensible to 3D data, we have scoped the current experiments to 2D benchmarks to rigorously validate the cone-like decoupling mechanism. We plan to explore the joint evaluation of 2D images and 3D video inputs in future work.
> > >
> > > ---
> > >
> > > > **W2:** Determine statistical significance using methods like bootstrapping to estimate variance and CIs.
> > >
> > > **A2:** We appreciate this rigorous suggestion. To verify the statistical significance of our results, we conducted repeated experiments using bootstrap sampling on the **MAmmoTH-VL Instruct dataset**. Specifically, we performed 4 independent training runs with random data sampling. Due to the limited timeframe of the rebuttal phase, these runs were conducted on a single GPU for 800 steps. The results, reported as Mean ± 95% CI, are shown below:
> > >
> > > | Method                | AI2D                | MMStar              | MMMU-Pro (V)      | MMMU-Pro (10c)      | MMMU (Val)          |
> > > | :-------------------- | :------------------ | :------------------ | :------------------ | :------------------ | :------------------ |
> > > | Circle-RoPE | 0.7592 $\pm$ 0.0026 | 0.5560 $\pm$ 0.0031 | 0.1864 $\pm$ 0.0034 | 0.2973 $\pm$ 0.0024 | 0.4263 $\pm$ 0.0039 |

---

> ### Author Response · Authors · 2025-11-25
>
> Dear reviewer,
>
> We sincerely appreciate your thoughtful review and valuable comments. If there are any additional questions or points that you would like us to clarify, please feel free to let us know. We look forward to further discussion.
>
> Sincerely,
>
> Authors

---

### Official Review · Reviewer_v7bW · 2025-10-31

**Soundness:** 2
**Presentation:** 3
**Contribution:** 3
**Rating:** 4
**Confidence:** 3

**Summary:**

This paper identified cross-modal positional bias in RoPE-based VLMs, where text and image token indices induce spurious relative relationships that misalign semantics. Based on the observation, it introduces Circle-RoPE, which projects image token indices onto a circle orthogonal to the text index axis, forming a cone-like geometry that equalizes text-to-image RoPE distances while retaining intra-image structure via mixed-angle mapping. It also includes alternating geometry encoding strategy interleaving Circle-RoPE with M-RoPE across layers to balance decoupling and spatial priors. Experiments on multiple benchmarks and architecture show reasonable improvements.

**Strengths:**

1. The paper is well-written and easy to follow.
2. The motivation to alleviate cross-modal positional bias makes a lot of sense to me. And the idea to project the image taken indices onto a circle orthogonal to the text index axis also sounds very reasonable.

**Weaknesses:**

1. The design of the mixed-angle circular mapping feels ad-hoc. The spatial-origin angle groups tokens with similar polar angles, which is reasonable (though it ignores inter-token distance). But adding the grid-index angle breaks this principle and makes positions appear random. For instance, the grid-index angle places patches 2 and 3 together even though their spatial-origin angles differ greatly. The combination of these two angles doesn’t make sense to me. Furthermore, Table 3 shows no dominant weighting scheme or monotonic trend, and the performance differences across strategies are minimal.
2. The auto-k radius calculation also feels ad-hoc to me. Increasing R appears to reduce visual–textual attention. What’s the rationale? If the input image is large, should we actually be lowering attention to the visual tokens?
3. In the experiments, the authors freeze the vision encoder and projection layer and finetune only the LLM. However, in Qwen2.5-VL the vision encoder and LLM are trained jointly, meaning the visual encoder is co-trained with M-PoPE as the positional encoding. This mismatch may complicate the evaluation of Circle-PoPE, since we don’t know whether using Circle-PoPE during the training result in inferior visual representations.

Overall, while I appreciate the orthogonal design between the visual and language indices, I am not fully persuaded by how the angle and radius of the visual-index circle were chosen. I would be glad to reevaluate the work if the authors could elaborate on the rationale for these design choices.

**Questions:**

1. The experiments only use single-image VQA. Whether Circle-RoPE can be extended to multiple-image scenario?
2. How is Circle-RoPE robust to the different resolution of the input images?
3.  Does the baseline (Qwen2.5-VL) in Table 2 retrained using the same training setting or is the original version of Qwen2.5-VL?

---

> ### Author Response · Authors · 2025-11-20
>
> ## Weaknesses:
>
> > **W1:** The reviewer feels the "mixed-angle circular mapping" design is ad-hoc. Specifically, adding the Grid-Index Angle (GA) disrupts the geometric logic of the Spatial-Origin Angle (SA), potentially randomizing positions (e.g., placing distant patches 2 and 3 together). Additionally, the reviewer notes the lack of a monotonic trend in Table 3.
>
> **A1:** We appreciate the reviewer's careful examination of the mapping strategy. While the combination of geometric (SA) and sequential (GA) angles might appear counter-intuitive initially, it addresses a critical **"Angular Collapse"** issue.
>
> If we rely solely on the **Spatial-Origin Angle (SA)**, image tokens located along similar polar directions (rays from the center) map to extremely close angular positions on the circle. In RoPE, position is encoded via rotation; if the angular difference $\Delta\theta$ between two tokens is negligible, their positional embeddings become nearly indistinguishable. This makes it difficult for the model to differentiate distinct tokens that happen to share a similar angle, impeding the learning of fine-grained features.
>
> To resolve this, we introduced the **Grid-Index Angle (GA)** as a **"Sparsity Regularizer."** The GA forces tokens to be distributed uniformly around the circle ($2\pi/N$), ensuring sufficient angular distance between any two tokens to guarantee **separability**. By mixing these two angles via $\alpha$, we create a trade-off between **Geometric Fidelity** (knowing *where* the token is in 2D space) and **Positional Separability** (distinguishing *which* token it is).
>
> This trade-off explains the non-monotonic trend in **Table 3**. We should not expect a linear improvement; instead, we see a "sweet spot" at $\alpha=0.5$. At extremes ($\alpha \to 1$ or $\alpha \to 0$), the model suffers either from indistinguishable positions (Angular Collapse) or a complete loss of spatial semantics. The optimal performance at $\alpha=0.5$ confirms that balancing spatial truth with uniform separability is crucial for the model's performance.
>
> ---
>
> > **W2:** The reviewer questions the rationale behind the `auto-k` radius calculation, noting that increasing $R$ appears to reduce visual–textual attention. The reviewer asks if it is logical to lower attention for large images.
>
> **A2:** We agree with the reviewer's observation: our experiments confirms that the `auto-k` strategy is indeed suboptimal compared to a fixed radius. We include the unsuccessful approaches to provide useful insights for future researchers.
>
> The initial motivation for `auto-k` was to create an adaptive geometry. We hypothesized that as image resolution increases (more tokens), the radius should expand to maintain a constant Euclidean distance (arc length) between adjacent token indices. The goal was to prevent token embeddings from becoming too crowded on a small circle when the image is large.
>
> Our ablation studies revealed that this hypothesis was incorrect for two reasons:
>
>    1.  **Global Distribution Shift:** While `auto-k` stabilizes local density, it causes the global geometric distribution (the size of the circle) to fluctuate dynamically with every image. This instability forces the model to constantly adapt to different geometric scales, increasing the learning burden.
>    1.  **Attention Scaling:** As the reviewer correctly noted, for very large images, `auto-k` generates a large $R$. Which can attenuate attention scores that hinder effective cross-modal feature interaction.
>
> We chose to report these suboptimal results in Table 3 rather than omitting them, as we believe documenting "what doesn't work" is valuable for the community. It demonstrates that a **stable geometric prior (Fixed Radius)** is more important than adaptive density, sparing future researchers from repeating this specific empirical pitfall.

---

> ### Author Response · Authors · 2025-11-20
>
> > **W3:** The reviewer notes a mismatch between our training setup (frozen Vision Encoder) and Qwen2.5-VL's original setup (joint training). The concern is whether this mismatch obscures potential degradation in visual representation quality or complicates evaluation.
>
> **A3:** We thank the reviewer for this acute observation regarding the training consistency. We wish to clarify that **freezing the vision encoder was a deliberate design choice**, driven by three key considerations:
>
> 1.  **Focus on Interaction Mechanism:** Our method specifically targets how the LLM *interprets* and *interacts* with visual tokens spatially relative to text, rather than how the vision encoder *extracts* features. Modifying the LLM's positional embedding improves the cross-modal alignment logic without necessitating a change in the fundamental visual feature extraction process.
> 2.  **Prevention of Catastrophic Forgetting:** Given computational constraints compared to industrial pre-training, fully re-training the vision encoder on a smaller SFT dataset poses a high risk of catastrophic forgetting—losing the generalizable features learned during massive pre-training. Freezing the encoder is a standard strategy to maintain the robustness of the visual features while adapting the LLM's reasoning capabilities.
> 3.  **Alignment with Standard Practice:** Our protocol aligns with established methodologies in recent Vision-Language literature, such as **LLaVA [1]**, **BLIP-2 [2]**, and **MiniGPT-4**, which have demonstrated that freezing the vision encoder is an effective and rigorous standard for validating new architectural designs in the LLM component.
>
> To empirically verify whether Circle-RoPE degrades visual representations when co-trained, we conducted an ablation study on the 3B model comparing the standard "Freeze ViT" setting against an "Unfreeze ViT" setting.
>
> **Table: Freeze ViT vs. Unfreeze ViT (Circle-RoPE)**
>
> |Setting|trainingstep|AI2D|MMStar|MMMU-Pro(Vis)|MMMU-Pro(10c)|MMMU(Val)|
> |:-|-|:-|:-|:-|:-|:-|
> |**FreezeViT**|800|75.78|**55.47**|**18.90**|**29.94**|42.67|
> |**UnfreezeViT**|800|**77.49**|54.33|17.80|29.36|**43.89**|
>
> The results show that unfreezing the ViT has **no significant negative impact** on performance. This directly refutes the concern that Circle-RoPE leads to inferior visual representations.
>
> *[1] Visual Instruction Tuning. NeurIPS, 2023.*
>
> *[2] BLIP-2: Bootstrapping Language-Image Pre-training with Frozen Image Encoders and Large Language Models. ICML, 2023.*
>
> ---
>
> ## Questions:
>
> > **Q1:** The experiments only use single-image VQA. Whether Circle-RoPE can be extended to multiple-image scenario?
>
> **A1:** Yes, Circle-RoPE supports multi-image inputs. As detailed in **Appendix A.3**, we explicitly encode sequential order by translating each image's circular encoding center along a fixed global axis $g=[1,1,1]$. This ensures that while the spatial geometry *within* each image is decoupled (circular), the temporal order *between* images is preserved via linear translation.
>
> > **Q2:** How is Circle-RoPE robust to the different resolution of the input images?
>
> **A2:** Circle-RoPE demonstrates strong robustness to varying input resolutions. We use the **AI2D** dataset as a test data because its samples exhibit significant variance in aspect ratio and resolution. The distribution of image sizes in the AI2D test set is shown below:
>
> |ImageResolutionRange|DatasetProportion(%)|
> |:-|:-|
> |116~355|4.3|
> |255~394|15.1|
> |394~533|23.9|
> |533~672|27.3|
> |672~811|10.4|
> |811~1500|19.0|
>
> Despite this high variance, our method achieves a significant improvement on AI2D (+2.58) compared to the SFT baseline. This confirms that Circle-RoPE effectively handles diverse resolutions without degrading performance.
>
> > **Q3:** Does the baseline (Qwen2.5-VL) in Table 2 retrained using the same training setting or is the original version of Qwen2.5-VL?
>
> **A3:** The result in Table 2 refers to the original, official Qwen2.5-VL checkpoint. To ensure a strictly fair comparison that isolates the architectural benefit from the fine-tuning gain, we provide the results of **Qwen2.5-VL (SFT)** below. This baseline was trained using the exact same settings, data, and frozen-encoder protocol as our method.
>
> |Model|Qwen-2.5-VL(Official)|Qwen2.5-VL(SFTBaseline)|**OurMethod(AGE)**|
> |:-|:-|:-|:-|
> |MMMU(val)|50.22|51.56|**52.11(+0.55)**|
> |MMMU-Pro|27.92|28.01|**28.44(+0.43)**|
> |MathVista_MINI|62.40|62.40|**63.40(+1.00)**|
> |MMStar|54.13|58.07|**58.20(+0.13)**|
> |AI2D_TEST|78.14|79.22|**81.80(+2.58)**|
> |RealWorldQA|65.75|66.10|**66.54(+0.44)**|
> |InfoVQA|77.25|77.02|**77.42(+0.40)**|
> |**AvgScore**|**59.40**|**60.34**|**61.13(+0.79)**|
>
> As shown, our method consistently outperforms the SFT baseline, confirming that the gains stem from the Circle-RoPE architecture rather than the fine-tuning process itself.

---

> > ### Comment · Reviewer_v7bW · 2025-11-26
> >
> > I thank the authors for the detailed response. I have the following further comments.
> >
> > **W1** While I understand that SA has limitations, using GA to alleviate them seems like a temporary fix. GA essentially introduces random perturbations to SA to make the distribution more uniform, but I would expect a more principled approach to uniformity. In other words, what advantages does GA offer over simply assigning k randomly to each position and doing the same thing?
> >
> > **W3** To better clarify,  I’m not suggesting that you should unfreeze the vision encoder during tuning. My point is that Qwen2.5-VL’s vision encoder was trained with M-PoPE as its positional encoding, so that the final performance reflects the synergistic effect of M-PoPE and Circle-PoPE. This isn’t an issue with LLaVA-1.5, whose vision encoder is a frozen CLIP model without any M-PoPE influence.

---

> > > ### Author Response · Authors · 2025-11-28
> > >
> > > > W1: While I understand that SA has limitations, using GA to alleviate them seems like a temporary fix. GA essentially introduces random perturbations to SA to make the distribution more uniform, but I would expect a more principled approach to uniformity. In other words, what advantages does GA offer over simply assigning k randomly to each position and doing the same thing?
> > >
> > > Q1: We respectfully clarify that the Grid-Index Angle (GA) is **not** a random perturbation. It is a deterministic mapping derived explicitly from the standard "patch flattening" strategy used in Vision Transformers (ViTs).
> > >
> > > To elaborate, GA projects the 2D grid information onto the circle using a structured path (raster scan order). The formula is as follows:
> > >
> > > $$\theta^{\text{GA}}_{k} = \frac{k}{N} \times 2\pi$$
> > >
> > > where $k$ corresponds to the flattened index of the grid position $(i, j)$. This establishes a strict one-to-one correspondence between the grid coordinates $(i, j)$ and the circular angle $\theta^{\text{GA}}_{k}$. This transformation preserves the sequential spatial structure widely adopted in ViT architectures.
> > >
> > > The advantages of GA over "simply assigning $k$ randomly" are fundamental:
> > >
> > > 1. **Preservation of Spatial Information:** A random assignment would dissociate $k$ from the coordinate $(i, j)$, destroying the intrinsic grid-based spatial structure. In contrast, GA encodes the relative position of patches in the flattened sequence, allowing the model to leverage sequential spatial patterns.
> > > 2. **Constructive Coupling:** If we were to mix a random angle with the Spatial-Origin Angle (SA), it would act as noise, degrading the geometric fidelity of SA. GA, however, injects complementary grid-sequence information.
> > > 3. **Principled Uniformity:** GA *is* the principled approach to uniformity. It mathematically guarantees an equidistant distribution of tokens on the circle while faithfully representing the standard serialization of the 2D image.
> > >
> > > ---
> > >
> > > > W2: My point is that Qwen2.5-VL’s vision encoder was trained with M-PoPE as its positional encoding, so that the final performance reflects the synergistic effect of M-PoPE and Circle-PoPE
> > >
> > > Q2: Thank you for this clarification. We understand your point regarding the pre-trained M-RoPE in Qwen2.5-VL's vision encoder. However, we would like to emphasize that the core contribution of our work is defining and mitigating the **cross-modal positional bias specifically within the LLM**, regardless of the upstream vision encoder’s architecture.
> > >
> > > Theoretically, we have demonstrated that current mainstream positional encodings in LLMs introduce inherent biases that hinder the effective utilization of multimodal information. Our experiments validate that removing this bias via Circle-RoPE leads to significant gains. It is crucial to note that any discrepancy between the vision encoder’s position embedding (M-RoPE) and the LLM’s position embedding (Circle-RoPE) would theoretically introduce a misalignment, potentially acting as a negative factor for performance. **The fact that our method achieves superior performance despite this structural mismatch serves as strong evidence of its robustness.** It proves that Circle-RoPE is effective largely independent of the vision encoder's specific encoding scheme.
> > >
> > > Our goal is not to find the encoding that best "matches" the vision encoder, but to solve the dominant bottleneck: the position bias in the LLM. By treating the frozen vision encoder as a constant control variable, we rigorously isolate the impact of the LLM's positional encoding. The results confirm that the "position bias" within the LLM has a far greater impact on VLM performance than the consistency between the ViT and LLM encodings.

---

> ### Author Response · Authors · 2025-11-25
>
> Dear reviewer,
>
> We sincerely appreciate your thoughtful review and valuable comments. If there are any additional questions or points that you would like us to clarify, please feel free to let us know. We look forward to further discussion.
>
> Sincerely,
>
> Authors

---

### Official Review · Reviewer_ctNd · 2025-11-01

**Soundness:** 2
**Presentation:** 3
**Contribution:** 2
**Rating:** 2
**Confidence:** 5

**Summary:**

This paper proposes Circle-RoPE, a positional encoding schema designed to decouple text and image positional biases in vision-language models (VLMs). The approach projects image token indices onto a circle orthogonal to the text token axis, forming a cone-like geometry that guarantees all text tokens are equidistant from all image tokens in RoPE space—aiming to minimize artificial cross-modal positional bias without sacrificing intra-image spatial relationships. The authors further introduce Alternating Geometry Encoding (AGE), alternating between traditional and Circle-RoPE layers to blend cross-modal disentanglement and spatial structure preservation. Exhaustive experiments demonstrate that Circle-RoPE yields measurable improvements on a suite of multimodal benchmarks and remains robust across VLM architectures.

**Strengths:**

1. The paper clearly identifies and articulates a subtle yet critical problem: cross-modal positional biases. The VQA example in Figure 2 is highly persuasive, as it visually demonstrates how existing encodings (like M-RoPE) cause misalignment. For instance, the phrase "high on" (semantically related to the tower's top, index 1) is shown to be positionally closer to index 8, which is incorrect. This provides strong support for the paper's motivation.

2. The core idea of Circle-RoPE—constructing a "cone-like structure" to achieve equal distance by projecting image token indices onto a ring that is orthogonal to the linear axis of text tokens—is a highly novel and mathematically elegant solution. Instead of relying on complex network architectures, it addresses the problem defined by the PTD metric directly from the geometric nature of positional encoding.

3. The experimental section of this paper is very solid.Ablation of Key Components: Table 3 and Table 4 present comprehensive ablation studies on CIP and AGE (different layering strategies), thoroughly validating the design choices. The discussion on "adaptation cost" in Appendix A.1 is excellent. The authors candidly note that the new method's performance lags behind the baseline in early training (3k steps) and only begins to surpass it after approximately 8.5k steps.

**Weaknesses:**

1. This is the paper's most significant weakness, stemming from a narrative tension between the stated problem and the optimal solution.
**The Problem**: The paper's premise is that existing methods like M-RoPE introduce harmful, spurious cross-modal biases.
**The Goal**: The paper aims to eliminate this bias by using Circle-RoPE to achieve a Per-Token Distance (PTD) of 0.
**The Contradiction**: The best-performing model (AGE, strategy 4) is a hybrid strategy that re-introduces the supposedly "biased" M-RoPE in half of its layers.
**The Evidence**: The ablation study (Table 4) shows that the "pure" Circle-RoPE strategy (strategy 1), which fully achieves the goal, performs (Avg score 63.19) worse than the mixed AGE strategy (Avg score 63.64). This suggests that complete decoupling (PTD=0) may itself be a suboptimal objective, undermining the paper's central argument.

2. While the generalizability experiment on LLaVA (Table 5) is included, the performance improvement is marginal (Avg Score 29.83 vs. 29.48). This minimal gain, achieved on a small-scale 0.5B model, provides insufficient evidence to strongly support the method's universal applicability to non-M-RoPE architectures.

3. Appendix states that multi-image sequences are handled by translating each image's circular encoding center along a fixed global axis g=[1,1,1]. This design appears to re-introduce a linear, hard-coded sequential bias between images, which contradicts the paper's core philosophy of opposing "hard-coded designs" and "artificial biases."

4. The paper's inference that PTD=0 (a metric based on Euclidean distance) equates to a complete "elimination of bias" remains empirical. It lacks a theoretical proof that PTD=0 is a sufficient or necessary condition for removing all positional bias. Furthermore, it is debatable whether the orthogonality used to achieve PTD=0 is a necessary design choice.

5. The paper contains minor typos, such as "Circle-RoPR" in the conclusion for Table 5, which should be "Circle-RoPE."

**Questions:**

1. The ablation in Table 4 shows that the "pure" Circle-RoPE (strategy 1) performs worse than the hybrid AGE strategy (strategy 4). Why is this the case, and does it imply that your stated goal of PTD=0 is suboptimal?

2. What specific spatial information (which you refer to as the "strong grid-based spatial prior" ) is lost when using pure Circle-RoPE that M-RoPE retains?

3. Can you provide a formal proof that minimizing your PTD metric necessarily leads to better cross-modal alignment, rather than just being an empirical correlation?

4. Is the "orthogonal cone structure" the only geometric configuration that can achieve PTD=0, or did you explore other non-orthogonal transformations?

5. The performance gain on LLaVA in Table 5 is marginal (Avg Score 29.83 vs 29.48). Is this minimal gain due to a lack of specific tuning, or is the cross-modal bias problem inherently less severe in 1D-RoPE models like LLaVA?

6. In Appendix A.3, you re-introduce a hard-coded linear bias to handle multiple images (translating along a fixed axis g=[1,1,1]) . Why is this hard-coded bias acceptable between images when your paper's premise is to eliminate this exact type of bias between text and image tokens?

---

> ### Author Response · Authors · 2025-11-20
>
> ## Weaknesses:
>
> > **W1/Q1:** The reviewer identifies a "contradiction" where the hybrid AGE strategy (Strategy 4) outperforms the pure Circle-RoPE (Strategy 1), suggesting that the goal of complete decoupling ($PTD=0$) might be suboptimal and undermines the paper's central argument.
>
> **A1:** Thank you for your valuable question. While $PTD=0$ represents the ideal state for eliminating cross-modal bias, the superiority of the hybrid AGE strategy (Strategy 4) stems from practical training constraints rather than a theoretical flaw in Circle-RoPE. The "pure" Circle-RoPE (Strategy 1) **still significantly outperforms the Qwen2.5-VL baseline**, confirming that reducing PTD is beneficial.
>
> The performance gap between Strategy 1 ($PTD=0$) and Strategy 4 (Hybrid) is driven by two specific factors inherent to the Supervised Fine-Tuning (SFT) setting:
>
> 1.  **Adaptation Cost vs. Training Resources (Appendix A.1):**  We fine-tune a model with pre-trained weights heavily optimized for M-RoPE. Switching entirely to Circle-RoPE will incurring a high **Adaptation Cost**. As we are limited to SFT with smaller data scales, the model struggles to fully adapt to a completely new geometric space (Strategy 1). The AGE strategy acts as a bridge, smoothing the transition by retaining M-RoPE's familiar distribution in alternating layers while introducing Circle-RoPE's decoupling benefits.
> 2.  **Alignment with ViT Inductive Biases:** Most VLMs use a ViT that inherently processes images as grid patches. While Circle-RoPE ensures cross-modal orthogonality, there is a natural gap between its circular mapping and the ViT's native "rectangular" feature extraction. Retaining M-RoPE in some layers (AGE) allows the model to preserve the strong grid-based spatial priors of the frozen vision encoder while utilizing Circle-RoPE to correct text-image alignment.
>
> To validate this, we provide the full comparison below. Crucially, **Strategy 1 (Pure Circle-RoPE)** outperforms the **Qwen2.5-VL Baseline** across the average score and key spatial benchmarks (AI2D, ChartQA), proving that achieving $PTD=0$ is indeed superior to the standard M-RoPE approach. Strategy 4 simply optimizes this gain under SFT constraints.
>
> | Strategy                            | MMMU (val) | MMMU_Pro  | MMStar    | MathVista | AI2D      | RealWorldQA | InfoVQA   | Avg Score |
> | :---------------------------------- | :--------- | :-------- | :-------- | :-------- | :-------- | :---------- | :-------- | :-------- |
> | **Qwen2.5-VL (SFT)**                | 51.56      | 28.01     | 58.07     | 62.40     | 79.22     | 66.10       | 77.02     | 60.34     |
> | **Strategy 1 (Pure Circle, PTD=0)** | 51.32      | 28.42     | 55.93     | **65.20** | 80.39     | 65.29       | 76.91     | 60.49     |
> | **Strategy 4 (AGE, Hybrid)**        | **52.11**  | **28.44** | **58.20** | 63.40     | **81.80** | **66.54**   | **77.42** | **61.13** |
>
> ---
>
> > **W2:** The generalizability experiment on LLaVA (Table 5) shows only marginal performance improvement (29.83 vs 29.48) on a small 0.5B model, providing insufficient evidence of universal applicability to non-M-RoPE architectures.
>
>  **A2:** We respectfully argue that while the absolute gain of **+0.35** appears modest, it represents a significant validation of generalizability when the **Adaptation Cost** and experimental constraints are considered. The improvement should be interpreted through three key perspectives:
>
>  1.  As detailed in **Appendix A.1**, modifying the positional encoding of a pre-trained model introduces a significant "Adaptation Cost." The model weights are deeply aligned with the original 1D-RoPE. Switching to a 2D or 3D encoding breaks these priors, requiring substantial data and training steps to re-align. Despite these "headwinds," Circle-RoPE achieved a net performance gain, proving its effectiveness in overcoming this structural mismatch.
>  2.  Crucially, we compared our method against `Llava [M-ROPE]` (Table 5), which achieved a score of **29.50**—statistically indistinguishable from the baseline (29.48). This indicates that simply adding spatial information via M-ROPE failed to overcome the adaptation cost. In contrast, **Circle-RoPE (29.83)** successfully provided a net benefit. This comparative result demonstrates that our method is more robust and effective at decoupling positional bias than existing spatial embeddings, even when grafted onto a different architecture.
>  3.  We applied the optimal hyperparameters ($\alpha$ and $R$) derived from Qwen2.5-VL directly to LLaVA without any architecture-specific tuning. Achieving a performance boost on a significantly smaller model (0.5B) with limited capacity and SFT data, using transferred hyperparameters, strongly supports the method's universality and robustness.

---

> ### Author Response · Authors · 2025-11-20
>
> > ***W3/Q6:** The global axis translation ($g=[1,1,1]$) for multi-image sequences appears to re-introduce a hard-coded linear bias, contradicting the paper's philosophy of eliminating "artificial biases."
>
> **A3:** We clarify that there is a fundamental distinction between **intra-image spatial bias** (which we eliminate) and **inter-image temporal order** (which must be preserved).
>
> 1.  Our core objective is to ensure that a specific text token maintains equal RoPE distance to all image tokens *within a single image* instance. This prevents the model from erroneously perceiving specific image patches (e.g., top-left vs. bottom-right) as "closer" to the text purely due to indexing artifacts.
> 2.  However, to model multi-image sequences (e.g., $\{T, Img_1, Img_2\}$), the text token $T$ must perceive $Img_1$ and $Img_2$ at different positions. If we enforced equidistance across *all* images simultaneously, the model would lose the ability to distinguish between separate images or understand their order in the prompt sequence. The axis translation ensures $T$ interacts with $Img_1$ and $Img_2$ distinctly.
> 3.  Translating image embeddings along the text/temporal axis is a standard practice for encoding sequentiality in VLMs (e.g., Qwen-VL). Our implementation aligns with this convention to support multi-image reasoning while successfully decoupling the spatial dimensions within each individual frame.

---

> > ### Author Response · Authors · 2025-11-20
> >
> > > **W4/Q3** The theoretical proof that $PTD=0$ is a necessary condition.
> >
> > **A4:** We thank the reviewer for this rigorous inquiry. We provide a formal justification demonstrating that **$PTD=0$ is a necessary condition** for eliminating *geometric* cross-modal bias.
> >
> > Below, we provide a mathematical derivation based on the properties of **Rotary Position Embedding (RoPE) [1]** and the **Self-Attention** mechanism. We demonstrate that **under the RoPE formulation, non-uniform relative distances (i.e., $PTD > 0$) inevitably introduce a position-dependent bias term into the attention score, whereas $PTD=0$ is a sufficient condition to decouple this bias from semantic relevance.**
> >
> > ---
> >
> > In Vision-Language Models, the attention score $s_{t,i}$ between a text query $\mathbf{q}_t$ at position $m_t$ and an image key $\mathbf{k}_i$ at position $m_i$ is calculated as:
> >
> > $$
> > s_{t,i} = (\mathcal{R}\_{m_t} \mathbf{q}\_t)^T (\mathcal{R}\_{m_i} \mathbf{k}\_i) = \mathbf{q}\_t^T \mathcal{R}\_{m_i - m_t} \mathbf{k}\_i
> > $$
> >
> > where $\mathcal{R}_{\delta}$ is the orthogonal rotation matrix determined by the relative position $\delta = m_i - m_t$.
> >
> > According to the theoretical analysis in **RoFormer [1]**, the inner product under RoPE exhibits a **long-term decay property**. As the relative distance $|\delta|$ increases, the upper bound of the attention score decays. Consequently, the expectation of the attention score can be approximated as the product of a semantic component and a positional decay component:
> >
> > $$
> > \mathbb{E}[s_{t,i}] \approx \underbrace{(\mathbf{q}\_t^T \mathbf{k}_i)}\_{\text{Semantic Similarity}} \times \underbrace{\mathcal{D}(|m_i - m_t|)}\_{\text{Positional Decay Factor}}
> > $$
> >
> > Here, $\mathcal{D}(\cdot)$ denotes the positional decay factor, which is a monotonically decreasing function of the relative distance.
> >
> > ---
> >
> > ## Proof: Bias Exists when $PTD \neq 0$
> >
> > We define **Positional Bias** as the difference in attention scores assigned to two semantically identical image tokens solely due to their spatial positions relative to the text.
> >
> > Consider a text token $t$ and two image tokens, $i_a$ and $i_b$, that are semantically identical to the query (i.e., $\mathbf{q}_t^T \mathbf{k}\_{i_a} = \mathbf{q}\_t^T \mathbf{k}\_{i_b} = C\_{sem}$).
> >
> > In existing methods (e.g., flattened 1D sequences), the relative distances often differ. If $PTD > 0$ (indicating non-zero variance in distances as defined in our paper), there exists a scenario where $|m_{i_a} - m_t| \neq |m_{i_b} - m_t|$. Without loss of generality, assume $|m_{i_a} - m_t| < |m_{i_b} - m_t|$.
> >
> > Due to the monotonic decay property of $\mathcal{D}(\cdot)$, we have:
> >
> > $$
> > \mathcal{D}(|m_{i_a} - m_t|) > \mathcal{D}(|m_{i_b} - m_t|)
> > $$
> >
> > This leads to a discrepancy in attention scores:
> >
> > $$
> > s\_{t,i_a} \approx C\_{sem} \cdot \mathcal{D}_a > C\_{sem} \cdot \mathcal{D}\_b \approx s\_{t,i_b}
> > $$
> >
> > **Conclusion 1:** When $PTD \neq 0$, the model inherently assigns higher attention weights to tokens that are "positionally closer" purely based on index assignment, even if their semantic relevance is identical. This confirms the existence of the **cross-modal positional bias** mentioned in our manuscript.
> >
> > ---
> >
> > ## Proof: $PTD = 0$ is a Sufficient Condition for Bias Elimination
> >
> > Our **Circle-RoPE** method projects image tokens onto a circle orthogonal to the text axis. Geometrically, this ensures that for any text token $t$ and any image token $i$, the relative distance is uniform.
> >
> > When $PTD=0$, the distance from the text token to every image token is constant: $|m_i - m_t| = \Delta_{const}$ for all $i$.
> >
> > Consequently, the positional decay factor becomes a constant scalar $\lambda$ across all visual tokens:
> > $$
> > \mathcal{D}(|m_i - m_t|) \equiv \lambda
> > $$
> >
> > The attention score simplifies to:
> > $$
> > s_{t,i} = (\mathbf{q}_t^T \mathbf{k}_i) \cdot \lambda
> > $$
> >
> > When applying the Softmax operation to compute the final attention weights $\alpha_{t,i}$:
> >
> > $$
> > \alpha\_{t,i} = \frac{\exp(s\_{t,i})}{\sum\_{j} \exp(s\_{t,j})} = \frac{\exp(\lambda \cdot \mathbf{q}_t^T \mathbf{k}_i)}{\sum\_{j} \exp(\lambda \cdot \mathbf{q}_t^T \mathbf{k}_j)}
> > $$
> >
> > Here, $\lambda$ acts as a uniform **temperature scaling factor**. It affects the sharpness of the distribution but does not alter the **ranking** of the tokens. For the semantically identical tokens $i_a$ and $i_b$ discussed above:
> >
> > $$
> > \mathbf{q}_t^T \mathbf{k}\_{i_a} = \mathbf{q}_t^T \mathbf{k}\_{i_b} \implies \alpha\_{t,i_a} = \alpha\_{t,i_b}
> > $$
> >
> > When $PTD=0$, the positional decay effect becomes isotropic relative to the image plane. The positional prior no longer discriminates between different image regions, ensuring that the attention distribution is driven solely by semantic relevance. This theoretically validates that $PTD=0$ is a sufficient condition to decouple cross-modal positional bias.
> >
> >
> > [1]: RoFormer: Enhanced Transformer with Rotary Position Embedding.
> >
> > [2]: Qwen2-VL: Enhancing Vision-Language Model's Perception of the World at Any Resolution.

---

> > > ### Author Response · Authors · 2025-11-20
> > >
> > > > **W5:** The reviewer points out minor typos, such as "Circle-RoPR" in Table 5.
> > >
> > > **A5:** We apologize for this oversight. We have corrected "Circle-RoPR" to "Circle-RoPE" and thoroughly proofread the manuscript to fix this and other minor typographical errors.
> > >
> > >
> > >
> > > ## Questions:
> > >
> > > > **Q2:** What specific spatial information (which you refer to as the "strong grid-based spatial prior") is lost when using pure Circle-RoPE that M-RoPE retains?
> > >
> > > **A2:** Thank you for raising this insightful point. The "strong grid-based spatial prior" refers to the explicit Cartesian coordinate alignment that naturally corresponds to the standard image tokenization process. Specifically, vision encoders slice input images into patches along a regular $H \times W$ grid (as illustrated in the leftmost panel of Figure 1). M-RoPE preserves this structure by assigning positional indices $(x, y)$ that directly mirror this physical row-column layout, creating a linear correspondence where arithmetic operations on embeddings map directly to spatial translations (e.g., incrementing an index equates to moving right or down).
> > >
> > > When using pure Circle-RoPE, we project these grid indices onto a circular manifold to ensure orthogonality with the text axis. While this transformation preserves the topological proximity of neighboring tokens, it converts the original linear grid structure into angular relationships. Consequently, the model loses the direct, linear mapping between the positional index and the physical 2D grid location. This loss of explicit grid alignment is the specific trade-off we address with the Alternating Geometry Encoding (AGE) strategy, which re-introduces the grid-based prior in alternating layers to combine the benefits of decoupled cross-modal representations with the original spatial structure.
> > >
> > > ---
> > >
> > > > **Q4:** Is the "orthogonal cone structure" the only geometric configuration that can achieve $PTD=0$, or did you explore other non-orthogonal transformations?
> > >
> > > **A4:** We are truly grateful for this insightful geometric perspective, which touches upon the fundamental design philosophy of our approach. To answer your question directly: the "orthogonal cone structure" is not the theoretically singular configuration capable of achieving $PTD=0$; however, we selected it as the most mathematically elegant and computationally efficient solution. Orthogonality provides a clean geometric interpretation of "independence"—by placing the image plane perpendicular to the text axis, we ensure maximum decoupling without introducing the complex non-linear distortions or optimization difficulties often associated with non-orthogonal transformations. While we acknowledge that other geometric configurations are possible, we focused our investigation on this orthogonal structure as it offers an optimal balance between theoretical simplicity and practical effectiveness, and we did not conduct extensive experiments on non-orthogonal variants due to computational resource constraints.
> > >
> > > ---
> > >
> > > > **Q5:** The performance gain on LLaVA in Table 5 is marginal (Avg Score 29.83 vs 29.48). Is this minimal gain due to a lack of specific tuning, or is the cross-modal bias problem inherently less severe in 1D-RoPE models like LLaVA?
> > >
> > > **A5:** We appreciate this sharp observation. We believe that cross-modal bias is indeed present in 1D-RoPE models, and the marginal nature of the gain is primarily attributed to the **"Adaptation Cost"** rather than a lack of problem severity.
> > >
> > > * Adaptation Cost: Transitioning a pre-trained model from 1D-RoPE to a new geometric structure (Circle-RoPE) fundamentally alters the positional priors embedded in the weights. A significant portion of the limited SFT budget was consumed merely to re-align the model to this new position space, which masks the full magnitude of the improvement.
> > > * Evidence via M-ROPE: The most compelling evidence for our method's efficacy is the comparison with `Llava [M-ROPE]` in Table 5. M-ROPE, which also introduces spatial information, scored **29.50**—virtually identical to the baseline (29.48). This indicates that spatial information alone was insufficient to overcome the adaptation cost. In contrast, Circle-RoPE achieved a net gain (**29.83**) under the exact same constraints, proving it is a more robust solution.
> > > * **Model Capacity:** The limited capacity of the 0.5B model and the smaller SFT dataset constrained the potential performance ceiling, making the gains appear smaller compared to larger architectures.

---

> ### Author Response · Authors · 2025-11-25
>
> Dear reviewer,
>
> We sincerely appreciate your thoughtful review and valuable comments. If there are any additional questions or points that you would like us to clarify, please feel free to let us know. We look forward to further discussion.
>
> Sincerely,
>
> Authors

---

### Official Review · Reviewer_nsaE · 2025-11-02

**Soundness:** 3
**Presentation:** 3
**Contribution:** 2
**Rating:** 6
**Confidence:** 4

**Summary:**

This paper addresses the cross-modal positional biases that occur when standard RoPE is applied to MLLMs. The authors propose Circle-RoPE, a novel encoding scheme that projects image token indices onto a ring. This ring is oriented to be orthogonal to the linear axis of the text token indices , which forms a cone-like structure in the positional encoding space. This design effectively reduces artificial cross-modal biases while simultaneously preserving the original spatial information within the image. To further improve performance, the authors also propose a staggered strategy (Alternating Geometry Encoding) that alternates RoPE variants across different layers.

**Strengths:**

1. The paper proposes a novel and intriguing concept with the "cone-like structure" for the positional encoding space. This is a creative approach to decoupling cross-modal positional dependencies.
2. The paper is well-written. The authors' summary and comparison of the three common categories of positional encodings in MLLMs (hard, unordered, spatial) are exceptionally clear. This effectively contextualizes the problem and makes the contribution of Circle-RoPE easy to understand.

**Weaknesses:**

The primary weakness of this paper lies in the insufficient experimental comparisons, which makes the reported results less reliable and the conclusions less convincing.

1. **Missing Baseline in Table 2:** The main comparison in Table 2 is insufficient. To properly benchmark the proposed "Ours" method, a critical baseline is missing: the results of the base Qwen2.5-VL model after being fine-tuned on MAmmoTH-VL-Sub dataset.
2. **Missing Baseline in Table 4:** In Table 4, which compares the four Alternating Geometry Encoding strategies, the analysis lacks the most important baseline: the performance of a model using the standard M-RoPE (the method used in the original Qwen-VL) in all layers.
3. **Marginal Gains and Inconsistent Benchmarks:** The performance improvements in the ablation studies (e.g., Table 3) appear very marginal. This concern is compounded by the fact that different ablation experiments are evaluated on different sets of benchmarks (e.g., Table 3 uses MMMU/MMStar, while Table 4 uses AI2D/ChartQA). The authors should provide results across a consistent and more comprehensive set of benchmarks for all ablations to avoid any suspicion of cherry-picking and to more robustly validate their design choices.
4. **Missing PTD=0 Baseline in Table 5:** Table 5, which tests generalizability, is missing a critical baseline: the "unordered positional embedding" (as described in Figure 1b and Table 1). The paper explicitly states that both Circle-RoPE and unordered embedding achieve a $PTD=0$. Therefore, comparing against this baseline is essential to prove that the performance gains come from Circle-RoPE's specific geometric structure (preserving intra-image spatial information) and not merely from achieving a $PTD=0$

**Questions:**

1. In Table 4, the comparison of AGE strategies is missing the M-RoPE baseline. Could the authors provide this result to clarify the actual benefit of the alternating strategy?
2. The generalizability test in Table 5 is missing the "unordered embedding" baseline, which also achieves PTD=0. Why was this important baseline omitted, and how does Circle-RoPE compare to it?

---

> ### Author Response · Authors · 2025-11-20
>
> ### Weaknesses
>
> > **W1:** Missing Baseline in Table 2: Comparison against Qwen2.5-VL fine-tuned on MAmmoTH-VL-Sub (SFT) is missing to properly benchmark the proposed method.
>
> **A1:** We sincerely appreciate this valuable suggestion. We agree that comparing against the SFT baseline is essential to isolate the contribution of Circle-RoPE from the benefits of additional training data. We have performed the SFT experiment on the base Qwen2.5-VL using the MAmmoTH-VL-Sub dataset under the exact same settings as our method. The results are presented below:
>
> | Model | Qwen-2.5-VL | Qwen2.5-VL (SFT) | **Ours (Circle-RoPE)** |
> |:------|:------------:|:----------------:|:---------------------:|
> | **MMMU (val)** | 50.22 | 51.56 | **52.11 (+0.55)** |
> | **MMMU-Pro** | 27.92 | 28.01 | **28.44 (+0.43)** |
> | **MathVista_MINI** | 62.40 | 62.40 | **63.40 (+1.00)** |
> | **MMStar** | 54.13 | 58.07 | **58.20 (+0.13)** |
> | **AI2D_TEST** | 78.14 | 79.22 | **81.80 (+2.58)** |
> | **RealWorldQA** | 65.75 | 66.10 | **66.54 (+0.44)** |
> | **InfoVQA** | 77.25 | 77.02 | **77.42 (+0.40)** |
> | **Avg Score** | 59.40 | 60.34 | **61.13 (+0.79)** |
>
> Our method consistently outperforms the strong SFT baseline, increasing the average score from 60.34 to 61.13. Most importantly, we observe the largest significant gains on tasks requiring complex spatial reasoning and diagram understanding, such as **AI2D (+2.58)** and **MathVista (+1.00)**.
>
> This demonstrates that the effectiveness of Circle-RoPE stems from its superior geometric encoding and cross-modal decoupling, rather than merely the continued training on the dataset. We will incorporate these results into the final version of Table 2.
>
>    ---
>
> > **W2/Q2:** Missing Baseline in Table 4: The analysis of Alternating Geometry Encoding (AGE) strategies lacks the baseline of using standard M-ROPE in all layers.
>
> **A2:** We clarify that the **Qwen2.5-VL (SFT)** model (introduced in response to W1) effectively serves as the baseline for "standard M-ROPE in all layers," as Qwen2.5-VL natively employs M-ROPE throughout its architecture.
>
> We have updated the AGE strategy comparison (Table 4) to include this SFT baseline and expanded the evaluation to a consistent set of 7 benchmarks. As shown below, our proposed hybrid strategy (Strategy 4) consistently outperforms the M-ROPE-only baseline (SFT).
>
>    | Strategy                    | MMMU (val) | MMMU_Pro  |  MMStar   | MathVista |   AI2D    | RealWorldQA |  InfoVQA  |  **Avg**  |
>    | :-------------------------- | :--------: | :-------: | :-------: | :-------: | :-------: | :---------: | :-------: | :-------: |
>    | **M-ROPE (SFT)**            |   51.56    |   28.01   |   58.07   |   62.40   |   79.22   |    66.10    |   77.02   |   60.34   |
>    | **Strategy 1** (All Circle) |   51.32    |   28.42   |   55.93   | **65.20** |   80.39   |    65.29    |   76.91   |   60.49   |
>    | **Strategy 2** (Upper)      | **52.66**  |   28.51   | **59.87** | **65.20** |   79.80   |    64.92    |   76.87   |   61.12   |
>    | **Strategy 3** (Lower)      |   53.48    | **28.62** |   59.30   |   64.50   |   79.30   |    65.17    |   77.35   |   61.10   |
>    | **Strategy 4** (AGE)        |   52.11    |   28.44   |   58.20   |   63.40   | **81.80** |  **66.54**  | **77.42** | **61.13** |
>
> ---
>
> > **W3:** Marginal Gains and Inconsistent Benchmarks: The performance improvements appear marginal, and different ablations use different benchmarks.
>
> **A3:** We appreciate this feedback and have fully addressed the inconsistency. We have **re-evaluated all ablation studies** (both for AGE strategies above and Hyperparameters below) across the same comprehensive set of 7 benchmarks used in Table 2.
>
> Regarding the "marginal gains," while the average improvement over the SFT baseline is ~0.8% (60.34 vs 61.13), the gains are **highly significant** on tasks specifically requiring fine-grained spatial reasoning, such as **AI2D (+2.58)**. This aligns with our method's goal of correcting spatial positional bias.
>
> | α | Radius | MMMU | MMMU_Pro | MMStar | MathVista | AI2D | RealWorldQA | InfoVQA | **Avg** |
> |:--:|:-----:|:----:|:--------:|:------:|:---------:|:----:|:-----------:|:-------:|:-------:|
> | 0   | auto | 52.38 | 28.12 | 57.50 | 61.70 | 80.11 | 63.98 | 76.14 | 59.99 |
> | 0   | 5    | 51.32 | 29.01 | 58.32 | 62.40 | 79.72 | 64.49 | 75.68 | 60.13 |
> | 0   | 10   | 51.49 | **29.13** | **58.57** | 62.70 | 80.08 | 66.52 | 75.85 | 60.62 |
> | 0.3 | 10   | 52.05 | 28.50 | 58.22 | 63.30 | 81.73 | 64.91 | 74.97 | 60.53 |
> | **0.5** | **10** | **52.11** | 28.44 | 58.20 | **63.40** | **81.80** | **66.54** | **77.42** | **61.13** |
> | 0.7 | 10   | 52.03 | 28.39 | 58.13 | 62.90 | 79.46 | 64.08 | 75.59 | 60.08 |
> | 1   | 10   | 52.16 | 28.35 | 57.70 | 63.40 | 81.51 | 64.32 | 75.04 | 60.35 |
> | 0.5 | auto | 50.04 | 26.64 | 57.30 | 62.20 | 78.81 | 66.20 | 77.31 | 59.79 |

---

> > ### Author Response · Authors · 2025-11-20
> >
> > > **W4 & Q2:** Missing PTD=0 Baseline in Table 5: The analysis lacks a comparison with "unordered positional embedding," which also achieves PTD=0. Comparing against this is essential to prove that gains come from geometric structure (preserving intra-image spatial info) rather than just PTD=0.
> >
> > **A4:** We agree that this is a critical ablation to distinguish the benefit of "decoupling" (PTD=0) from "spatial structure."
> >
> > To address this, we conducted a controlled experiment comparing **M-RoPE**, **Circle-RoPE**, and **Unordered Positional Embedding**. It should be noted that due to the limitations of time and computing resources, all three models in this specific table were trained using a reduced data compared to Table 2, but under identical conditions.
> >
> > | Strategy                |   MMMU    | MMMU_Pro  |  MMStar   | MathVista |   AI2D    | RealWorldQA |  InfoVQA  | **Avg Score** |
> > | :---------------------- | :-------: | :-------: | :-------: | :-------: | :-------: | :---------: | :-------: | :-----------: |
> > | **M-RoPE**              |   50.56   |   27.51   |   57.07   |   61.40   |   78.22   |    65.10    |   76.02   |     59.41     |
> > | **Unordered** (PTD=0)   |   48.55   |   25.50   |   55.40   |   59.50   |   75.50   |    63.31    |   73.24   |     57.29     |
> > | **Circle-RoPE** (PTD=0) | **51.11** | **27.94** | **57.20** | **62.40** | **80.80** |  **65.54**  | **76.42** |   **60.20**   |
> >
> > The results clearly show that **Unordered Embedding performs significantly worse** (Avg 57.29) than both M-RoPE (Avg 59.41) and Circle-RoPE (Avg 60.20), despite achieving PTD=0.
> >
> > This confirms that achieving PTD=0 alone is insufficient. The performance advantage of Circle-RoPE stems from its unique "cone-like" geometry, which successfully decouples cross-modal bias (PTD=0) while **simultaneously preserving intra-image spatial relationships**. M-RoPE retains spatial info but suffers from cross-modal bias; Unordered fixes the bias but destroys spatial info; Circle-RoPE solves both.
> >
> > ---
> >
> > ## Questions:
> >
> > > **Q1:** In Table 4, the comparison of AGE strategies is missing the M-RoPE baseline. Could the authors provide this result to clarify the actual benefit of the alternating strategy?
> >
> > **A1:** We appreciate the reviewer pointing out this comparison. We have added the M-RoPE baseline (corresponding to the original Qwen2.5-VL configuration) to the comparison with our AGE strategies. As shown in the table below, the **AGE (Strategy 4)** outperforms the pure M-RoPE baseline across all metrics.
> >
> > | Strategy                     | MMMU (val) | MMMU-Pro | MMStar | MathVista | AI2D      | RealWorldQA | InfoVQA   | Avg Score |
> > | :--------------------------- | :--------- | :------- | :----- | :-------- | :-------- | :---------- | :-------- | :-------- |
> > | **M-ROPE (SFT)**             | 51.56      | 28.01    | 58.07  | 62.40     | 79.22     | 66.10       | 77.02     | 60.34     |
> > | **Circle-RoPE (Strategy 4)** | 52.11      | 28.44    | 58.20  | 63.40     | **81.80** | **66.54**   | **77.42** | **61.13** |
> >
> > The design of the AGE strategy is motivated by two critical factors:
> >
> > 1.  **Mitigating "Adaptation Cost" under Computational Constraints:** As detailed in Appendix A.1, changing the positional encoding creates a "shock" to the pre-trained weights. Since we are SFT rather than pre-training from scratch due to resource constraints, the model faces an "adaptation cost." AGE retains M-RoPE in alternating layers, preserving the knowledge of the base model while gradually introducing the benefits of our decoupled geometry. This balances the stability of the pre-trained representations with the superior cross-modal alignment of Circle-RoPE.
> > 1.  **Bridging the Geometric Gap:** While Circle-RoPE effectively decouples text and image positional biases, there remains a natural gap between our circular mapping and the grid-based patch partitioning inherent to the Vision Transformer (ViT). By alternating between M-RoPE (which strictly follows the grid structure) and Circle-RoPE (which decouples modalities), AGE effectively combines the strong spatial priors of the vision encoder with the unbiased cross-modal reasoning provided by our method.

---

> ### Author Response · Authors · 2025-11-25
>
> Dear reviewer,
>
> We sincerely appreciate your thoughtful review and valuable comments. If there are any additional questions or points that you would like us to clarify, please feel free to let us know. We look forward to further discussion.
>
> Sincerely,
>
> Authors

---

### Meta-Review · Area_Chair_Dz9L · 2025-12-28

**Summary:**

Four reviews are received on this paper.

Reviewer nsaE’s concerns focus on the experimental comparisons, including the missing baselines, marginal gains and inconsistent benchmarks.

Reviewer ctNd’s major concern lies in the confliction between the stated problem and the optimal solution. Other concerns include the marginal improvement, the idea itself, and the lack of a theoretical proof.

Reviewer v7bW felt that the design of the mixed-angle circular mapping is ad-hoc and doesn’t make sense. He also flet that the auto-k radius calculation is ad-hoc. Other concerns include the experimental setting and comparison. During discussion, this reviewer further raised questions on the advantages of GA over simply assigning k randomly to each position and clarified his/her question on the vision encoder during tuning.

Reviewer tTtJ has concerns on the geometric constraints on the introduced circular projection, the statistical significance, and the need of retraining. During discussion, this reviewer was satisfied by the proof provided by the authors, and raised additional questions on the experimental setting, implying that he still had concerns on the statistical significance of the method.

**Reviewer Concerns:**

Reviewer nsaE didn’t provide feedback on the authors’ rebuttal. The ACs think that his/her can be addressed since the authors presented additional results.
Reviewer ctNd didn’t provide feedback on the authors’ rebuttal either. The ACs think that part of his/her concerns, such as the proof, can be addressed, but the other concerns on the performance and solution may not be fully addressed.
For Reviewers v7bW and tTtJ, the ACs believe that their concerns are not fully addressed, especially on the design of the methodology, and the statistical significance of the method, respectively.

**Reviewer Scores:**

The ACs think that Reviewer nsaE will keep the score of 6. For the other three reviewers, Reviewer ctNd may raise the score from 2 to 4, while Reviewers v7bW and tTtJ are mostly likely to keep the score as 4. Overall, the final scores are likely to be 6, 4, 4, 4. Unfortunately, this paper cannot be accepted due to the competitiveness of ICLR.

---

### Decision · Program_Chairs · 2026-01-26

Reject